# Spatial and temporal analysis of PCP protein dynamics during neural tube closure

Mitchell T Butler[†], John B Wallingford*

Department of Molecular Biosciences, University of Texas at Austin, Austin, United States

**Abstract** Planar cell polarity (PCP) controls convergent extension and axis elongation in all vertebrates. Although asymmetric localization of PCP proteins is central to their function, we understand little about PCP protein localization during convergent extension. Here, we use quantitative live imaging to simultaneously monitor cell intercalation behaviors and PCP protein dynamics in the *Xenopus laevis* neural plate epithelium. We observed asymmetric enrichment of PCP proteins, but more interestingly, we observed tight correlation of PCP protein enrichment with actomyosin-driven contractile behavior of cell-cell junctions. Moreover, we found that the turnover rates of junctional PCP proteins also correlated with the contractile behavior of individual junctions. All these dynamic relationships were disrupted when PCP signaling was manipulated. Together, these results provide a dynamic and quantitative view of PCP protein localization during convergent extension and suggest a complex and intimate link between the dynamic localization of core PCP proteins, actomyosin assembly, and polarized junction shrinking during cell intercalation in the closing vertebrate neural tube.

DOI: https://doi.org/10.7554/eLife.36456.001

*For correspondence:
wallingford@austin.utexas.edu

Present address: [†]UNC Lineberger Comprehensive Cancer Center, University of North Carolina at Chapel Hill, Chapel Hill, United States

Competing interests: The authors declare that no competing interests exist.

## Introduction

Convergent extension (CE) is the evolutionarily conserved morphogenetic engine that drives elongation of the body axis in animals ranging from insects to mammals (*Tada and Heisenberg, 2012*). Multiple cell behaviors can contribute to CE, but by far the most well-understood process is cell intercalation, by which cells rearrange in a polarized manner (*Walck-Shannon and Hardin, 2014*). Cell intercalation, in turn, is thought to be driven by multiple subcellular behaviors, including extension of mediolaterally directed cellular protrusions (e.g. [*Keller and Hardin, 1987*; *Shih and Keller, 1992*]) and active shrinkage of mediolaterally oriented cell-cell junctions (e.g. [*Bertet et al., 2004*; *Blankenship et al., 2006*]). More recent data suggest that these two subcellular behaviors likely work in concert (*Sun et al., 2017*; *Williams et al., 2014*). Understanding the molecular mechanisms governing protrusive activity and junction shrinking during cell intercalation will be essential to understanding convergent extension.

The junction shrinking mechanism for cell intercalation was initially described in *Drosophila* (*Bertet et al., 2004*; *Blankenship et al., 2006*) and was subsequently identified in both epithelial and mesenchymal cells in vertebrates (*Lienkamp et al., 2012*; *Nishimura et al., 2012*; *Shindo and Wallingford, 2014*; *Trichas et al., 2012*; *Williams et al., 2014*). In all tissues examined by live imaging, junction shrinkage is accompanied by pulsed actomyosin contractions that are restricted to or enriched at mediolaterally oriented cell-cell junctions and absent from or less common at the junctions perpendicular to the anterior-posterior axis (*Bertet et al., 2004*; *Blankenship et al., 2006*; *Shindo and Wallingford, 2014*; *Williams et al., 2014*). A major unresolved question concerns the

molecular mechanism by which actomyosin activity is restricted to specific cell-cell junctions during intercalation.

In *Drosophila*, pair-rule genes and Toll receptors are crucial regulators of polarized actomyosin (*Paré et al., 2014*; *Zallen and Wieschaus, 2004*), but homologous genes do not appear to be involved in vertebrates. Instead, the most well-characterized regulator of mediolateral cell intercalation in vertebrates is the Planar Cell Polarity (PCP) signaling system (*Butler and Wallingford, 2017*). Indeed, PCP signaling controls cell intercalation during gastrulation and neural tube closure in frogs, fish, and mice, and mutations in PCP genes are now a well-defined genetic risk factor for human neural tube birth defects (NTDs) (*De Marco et al., 2014*; *Juriloff and Harris, 2012*; *Wallingford et al., 2013*). Understanding PCP signaling is therefore a critical challenge in developmental cell biology.

One fundamental principle of PCP signaling is that cell polarity is imparted by asymmetric enrichment of the core PCP proteins (*Butler and Wallingford, 2017*; *Strutt and Strutt, 2009*). As shown first in *Drosophila*, the Prickle (Pk) and Van Gogh (Vangl) proteins act in concert on one side of the cell, and Dishevelled and Frizzled act on the complementary side (*Axelrod, 2001*; *Bastock et al., 2003*; *Strutt, 2001*; *Tree et al., 2002*). Interestingly, FRAP studies in *Drosophila* have shown that these patterns of enrichment are driven by planar polarization of the junctional turnover kinetics of PCP proteins, underscoring the dynamic nature of the PCP signaling system (*Strutt et al., 2011*). Similar patterns of enrichment and turnover have been reported in vertebrate epithelia (*Butler and Wallingford, 2015*; *Chien et al., 2015*; *Shi et al., 2016*), but less is known about PCP protein localization dynamics during cell intercalation.

For example, complementary, asymmetric domains of PCP protein enrichment have been described during vertebrate CE (*Ciruna et al., 2006*; *Jiang et al., 2005*; *McGreevy et al., 2015*; *Ossipova et al., 2015*; *Roszko et al., 2015*; *Yin et al., 2008*), but how PCP protein enrichment is coordinated in space and time with the subcellular behaviors that drive intercalation remains essentially unexplored. This gap in our knowledge is critical, because recent work demonstrates that PCP proteins are required for the junction shrinking behaviors that contribute critically to cell intercalation (*Lienkamp et al., 2012*; *Nishimura et al., 2012*; *Shindo and Wallingford, 2014*). Thus, there is a pressing need for a quantitative, dynamic picture of PCP protein localization as it relates both to subcellular behaviors involved in cell intercalation and to the actomyosin machinery that drives them.

To this end, we established methods for robust quantification of PCP protein localization in a living vertebrate neural plate as well as methods for correlating PCP protein dynamics with the subcellular behaviors that drive epithelial cell intercalation. Strikingly, we find that in addition to expected patterns of spatial asymmetry, PCP protein enrichment is tightly linked to cell-cell junction behavior: Prickle2 (Pk2) and Vangl2 were dynamically enriched specifically at shrinking cell-cell junctions and depleted from elongating junctions during cell intercalation. FRAP analysis revealed that these patterns of enrichment reflected differences in the kinetics of protein turnover at these sites. Moreover, Pk2 enrichment was temporally and spatially correlated with planar polarized oscillations of junctional actomyosin enrichment. Importantly, all these dynamic relationships were disrupted when PCP signaling was manipulated. Thus, our studies reveal an intimate link between the dynamic localization of core PCP proteins, actomyosin assembly, and polarized junction shrinking during cell intercalation of the closing vertebrate neural tube.

## Results

We characterized PCP protein dynamics in the neural plate of *Xenopus*, as studies in this animal consistently prefigure similar results in mammalian systems yet provide exceptional views of dynamic cell biological processes. Because Dishevelled and Frizzled function in both PCP and canonical Wnt signaling, their roles are more difficult to interrogate, so we focused instead on Prickle and Vangl, which act solely in PCP signaling and are required for CE and neural tube closure in vertebrates, including *Xenopus* (*Darken et al., 2002*; *Goto et al., 2005*; *Goto and Keller, 2002*; *Kibar et al., 2001*; *Takeuchi et al., 2003*). Previous work suggests that Prickle and Vangl localize to the anterior face of cells in the *Xenopus* neural plate (*Ossipova et al., 2015*), but we sought to establish a more robust quantification of this pattern as a foundation for the time-lapse studies described below.

We used mRNA injection to express fluorescent protein (FP) fusions to PCP proteins in Xenopus. Because overexpression of PCP proteins has a well-documented effect on PCP signaling, we used titration experiments to empirically determine the lowest possible dose of mRNA that allowed

imaging in the *Xenopus* neural plate. Under these conditions, GFP-Vangl2 displayed a strong asymmetric bias to anterior cell faces (*Figure 1A*), while at slightly higher doses of injected mRNA, low levels of Vangl2 could be observed at the posterior faces as well (not shown), consistent with previous reports (*Ossipova et al., 2015*; *Roszko et al., 2015*). Core PCP proteins are encoded by multigene families, and despite the reported role of Prickle1 in convergent extension, GFP-Pk1 did not display asymmetric enrichment in our hands (not shown). However, GFP-Prickle2 (*Butler and Wallingford, 2015*) was strongly enriched anteriorly (*Figure 1A*). GFP-Pk2 was also restricted to the apicolateral cell junctional regions, as expected (*Figure 1—figure supplement 1*). Knockdown experiments confirmed that Pk2 was physiologically relevant for convergent extension and neural tube closure in *Xenopus* (*Figure 1—figure supplement 2*). In double-labeling experiments, GFP-Vangl2 strongly co-localized with RFP-Pk2 (*Figure 1A*, *Video 1*).

We quantified these localization patterns at the population level by binning cell-cell junctions into two groups based on their orientation. To maintain a consistent nomenclature with previous work on cell intercalation, we refer to junctions aligned within 45 degrees of the mediolateral axis as 'V-junctions' and the perpendicular junctions (between 45 and 90 degrees off the mediolateral axis) as 'T-junctions.' An important subtlety to note here is that the V-*junctions* are *mediolaterally* aligned, but actually separate *cells* that are *anteroposterior* neighbors. We found that GFP-Pk2 was significantly enriched at V-junctions as compared to T-junctions (*Figure 1B*). In a more granular view of the data, we observed a significant correlation between GFP-Pk2 pixel intensity and junction angle, with higher enrichment along more mediolaterally oriented junctions (*Figure 1C*).

To test these quantification schemes, we took advantage of the fact that in many other systems, disruption of any one core PCP protein leads to loss of polarized enrichment of the others. We used targeted injection to co-express reagents for disrupting PCP signaling together with a nuclear RFP lineage tracer, allowing us to compare normal and experimental cells in the same embryo (*Figure 1D,E*). Xdd1 is a well-defined, PCP-specific dominant negative of Dvl2 (*Sokol, 1996*; *Wallingford et al., 2000*) that disrupts convergent extension of the neural tube in *Xenopus* (*Wallingford and Harland, 2001*; *Wallingford and Harland, 2002*); we found that Xdd1 expression severely disrupted Prickle2 asymmetries in the neural plate using both ensemble and individual metrics (*Figure 1B,C*).

To extend this analysis, we explored Pk2 signaling by expressing a deletion construct lacking the PET and LIM domains (Pk2-ΔPΔL), as this construct disrupted PCP in *Xenopus* multiciliated cells and an equivalent deletion of Pk1 disrupts CE (*Butler and Wallingford, 2015*; *Takeuchi et al., 2003*). We found that Pk2-ΔPΔL strongly disrupted axis elongation in *Xenopus* (*Figure 1—figure supplement 3*) and also severely disrupted the planar asymmetry of co-expressed wild-type Pk2-GFP in the neural plate (*Figure 1B,C*). These results establish the *Xenopus* neural plate as an effective, quantitative platform for studies of PCP protein localization.

## Epithelial convergent extension in the closing *Xenopus* neural tube involves PCP-dependent polarized junction shrinking

Convergent extension is an inherently dynamic process, as cells constantly exchange neighbors. The dynamic nature of tissues engaged in convergent extension differs markedly from the settings in which PCP protein localization is most commonly studied. We therefore sought to exploit the strengths of *Xenopus* embryos to image PCP protein dynamics together with the subcellular behaviors that drive convergent extension in the closing neural tube. Curiously, the neural plate of *Xenopus* consists of two cell layers, an outer epithelial layer and a deeper mesenchymal layer (*Schroeder, 1970*). While cell intercalation of the deep mesenchymal cells has been characterized (*Elul et al., 1997*; *Keller et al., 1992*), the behaviors of the overlying epithelial cells have not. This distinction is not trivial, because it is the outer epithelial cells that display the robust patterns of PCP protein localization described above. To understand how PCP protein localization relates to convergent extension cell behaviors, we had first to characterize cell behaviors in this epithelium.

Neural tube closure spans roughly 6 hr in *Xenopus,* starting with the shaping of the neural tube at last gastrula stages, followed by the progressive elevation and apposition of the neural folds (*Figure 2A*). We immobilized embryos on a confocal microscope stage (*Kieserman et al., 2010*) and used image tiling to collect high-magnification images of the superficial neural plate across the entirety of neural tube closure (*Video 2*). This approach allowed assessment of tissue-level morphogenetic changes in the neural plate, as well as individual cell trajectories, which closely resembled

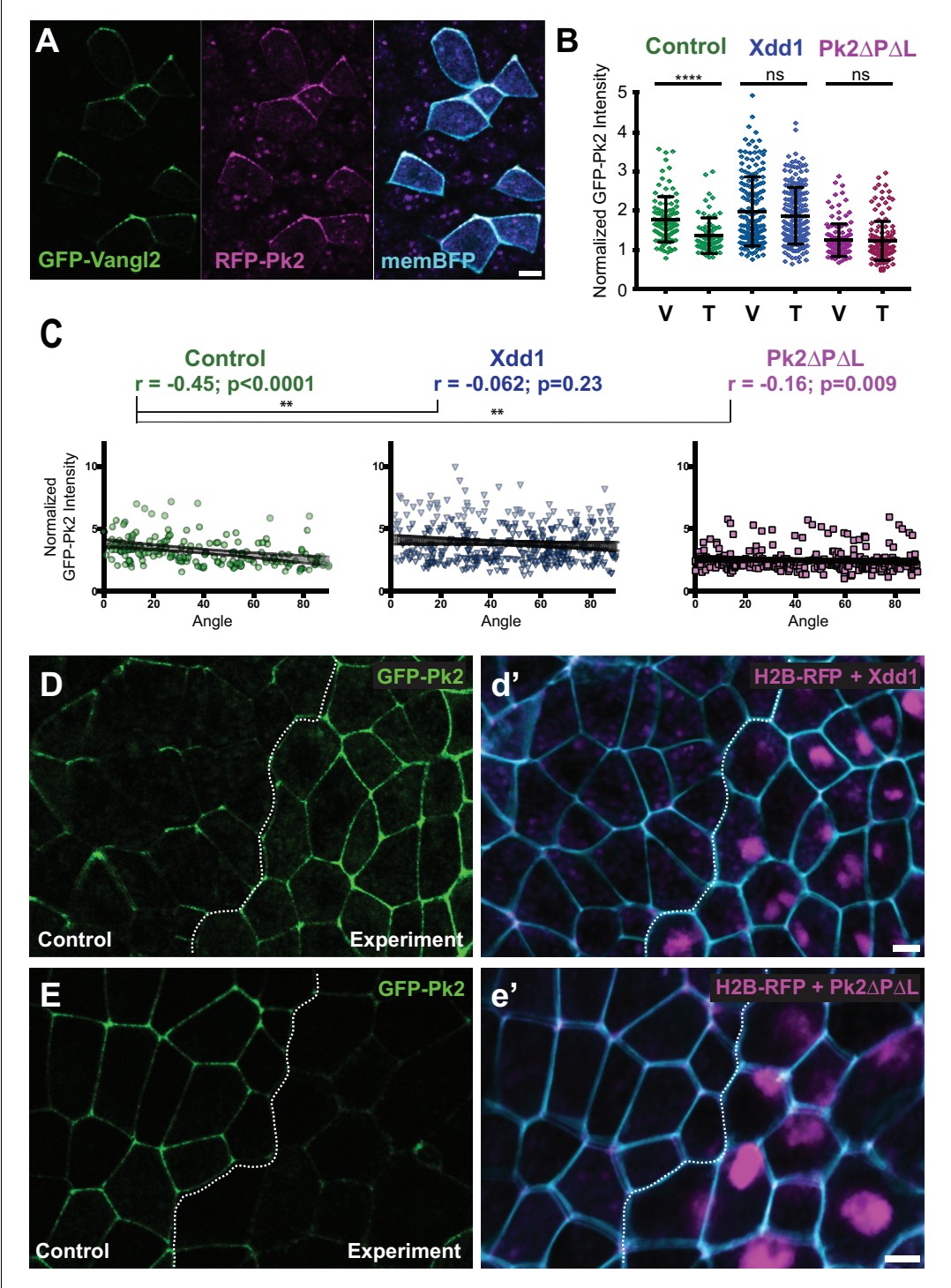

**Figure 1.** Planar polarized localization of Prickle2 and Vangl2 in the neural plate. (**A**) Neural epithelium mosaically labeled with GFP-Vangl2, RFP-Pk2, and membraneBFP showing the overlapping localizations of Pk2 and Vangl2. Anterior is up, and scale = 10 µm. (**B**) Graph plotting GFP-Pk2 intensity along V-junctions (0–45° relative to mediolateral axis) and T-junctions (46–90° relative to mediolateral axis) normalized as a ratio to the mean cytoplasmic intensity in control cells and cells expressing Xdd1 and Pk2-ΔPΔL. Error bars represent standard deviation. Ctrl V vs. T, p<0.0001***; Xdd1 V vs. T, p=0.5799; Pk2-ΔPΔL V vs. T, p=0.173; Ctrl V vs. Xdd1 T, p=0.5770; Ctrl T vs. Pk2-ΔPΔL V, p=0.0268 (Mann Whitney test for significance). n = 101 V and 71 T from three experiments, seven embryos (Ctrl); n = n = 171 V and 199 T from four experiments, five embryos (Xdd1); n = 128 V and 142 T from three experiments, seven embryos (Pk2-ΔPΔL). (**C**) Distributions of data shown in (**B**) plotted against the angle of the junction at which the intensity was measured. Correlation coefficients for Xdd1 and Pk2-ΔPΔL were significantly different from controls using the Fischer R-to-Z transformation. n = 172

*Figure 1 continued on next page*

*Figure 1 continued*

junctions (Control), n = 263 junctions (Xdd1) and n = 245 junctions (Pk2-ΔPΔL) (D-E) Confocal images of *Xenopus* neural epithelia labeled evenly with GFP-Pk2 and membraneBFP and mosaically with H2B-RFP, serving as a tracer for either Xdd1 (C) or Pk2-ΔPΔL (D) expression. Scale = 10 μm.

DOI: https://doi.org/10.7554/eLife.36456.002

The following figure supplements are available for figure 1:

**Figure supplement 1.** GFP-Pk2 localizes to anterior apicolateral regions of cells in the *Xenopus* neural plate.

DOI: https://doi.org/10.7554/eLife.36456.003

**Figure supplement 2.** Pk2 knockdown results in embryonic convergent extension phenotypes.

DOI: https://doi.org/10.7554/eLife.36456.004

**Figure supplement 3.** A dominant negative Pk2 disrupts convergent extension.

DOI: https://doi.org/10.7554/eLife.36456.005

those of previous studies ((*Figure 2B*; *Figure 2—figure supplement 1*) and see [*Keller et al., 1992*]). Moreover, our tiling approach provided sufficient magnification to quantify the discrete behaviors of individual cells, and tracking of cell clusters revealed extensive intercalations that were mediolaterally biased, resulting in convergence and extension (*Figure 2C*, *Video 3*).

Cell intercalations in the neural plate were associated with so-called T transitions, which are characterized by preferential contraction of junctions aligned in the mediolateral axis (T1-T2 transition) (*Figure 3A,a'*), followed by elongation of new junctions perpendicularly along the anteroposterior axis (T2-T3 transition)(*Figure 3a', a''*)(*Bertet et al., 2004*). As above, we first quantified these behaviors at the ensemble level and found that mediolaterally oriented V-junctions preferentially shrank, while the perpendicular T-junctions elongated (*Figure 3B*). Moreover, when we plotted the change in the length of cell-cell junctions against the average angle of that junction for all cells examined (n > 250), we observed a significantly positive correlation; V-junctions preferentially shrank along the mediolateral axis and T-junctions elongated perpendicularly (*Figure 3C*). Importantly, the orientation of shrinking and growing junctions in this analysis remained fairly constant, changing on average only 5 (±4) degrees over the course of measurement, and no junctions shifted by more than 20 degrees. Thus, in general, V-junctions separate anteroposterior (AP) neighbors and T-junctions separate mediolateral (ML neighbors). Finally, we also observed planar polarized formation and resolution of multicellular rosettes (*Figure 3D–F*), as have been described in other epithelia (*Blankenship et al., 2006*; *Lienkamp et al., 2012*; *Trichas et al., 2012*).

Because PCP signaling is essential for neural convergent extension, we next assessed the effect of PCP disruption specifically on junction shrinking behaviors in the neural epithelium. Using the mosaic approach described above, we found that expression of Xdd1 elicited the expected tissue level defect in the medially directed movement of the neural folds (*Figure 4A*, magenta nuclei indicate Xdd1 expressing cells).

Analysis of individual cell behaviors in these embryos revealed that Xdd1 expression significantly disrupted planar polarized junction shrinking, eliminating the strong correlation between junction angle and junction shrinkage or growth (*Figure 4B*, compare with *Figure 3C*). This uncoupling of junction behavior from orientation was associated with reduced junction shrinkage generally, and a reduction in the number of productive T transitions (*Figure 4C*). In fact, even the few productive T transitions that were observed were significantly slowed (*Figure 4D*).

Finally, these defects in cell intercalation behavior had a profound impact on cell shape. During neural plate shaping stages (12-14), the majority of neural epithelial cells in control embryos exhibited a shift from an anteroposterior orientation to a mediolateral orientation (*Figure 4E*, green), while Xdd1 expressing cells maintain their alignment in the anteroposterior

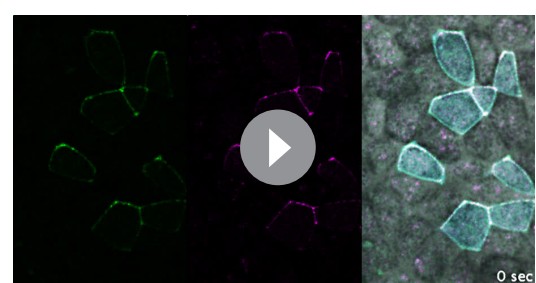

**Video 1.** Time-lapse confocal images of a *Xenopus laevis* neural plate mosaically labeled with GFP-Vangl2 (left), RFP-Pk2 (middle), and memBFP (right, merged with GFP-Vangl2, RFP-Pk2, and DIC channels). Still images and scale are shown in *Figure 1A*.

DOI: https://doi.org/10.7554/eLife.36456.006

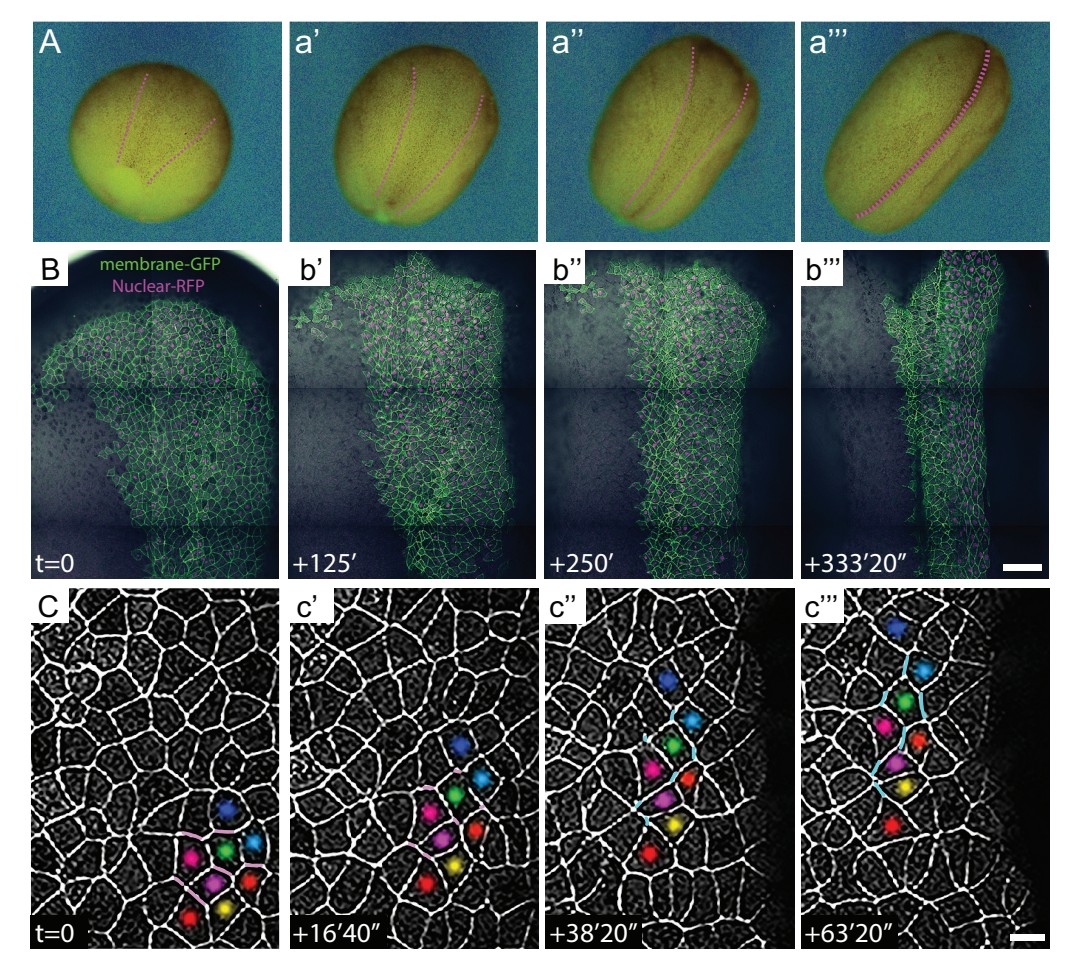

**Figure 2.** High-magnification time-lapse imaging of convergent extension in the closing *Xenopus* neural tube. (**A**) Stereo image stills from a time-lapse movie of *Xenopus* neural tube morphogenesis from stages 12 to 19. (**B**) Stills from time-lapse confocal imaging of the dorsal side of an embryo from stages 12 to 16. Cells labeled with membraneGFP and nuclear H2B-RFP are merged with the DIC image of the same embryo. Images are shown ~2 hr. apart (128 min. interval). Scale = 200 μm. (**C**) Higher magnification images of cell rearrangements in neural ectoderm from the time-lapse shown in (**B**). Labeling of individual cells with colored dots across time points demonstrates the cellular rearrangements contributing to the narrowing and lengthening of tissue and T transitions labeled with magenta for shrinking (T1-T2) and cyan for growing (T2-T3) junctions. Scale = approx. 20 μm.
DOI: https://doi.org/10.7554/eLife.36456.007

The following figure supplement is available for figure 2:

**Figure supplement 1.** In vivo imaging of cell trajectories in the closing neural tube.
DOI: https://doi.org/10.7554/eLife.36456.008

axis (*Figure 4E*, blue). In fact, these cells actually increased their length-to-width ratios along the AP axis (not shown), likely as a result of forces generated by other cell behaviors that shape the neural plate at these stages (e.g. apical constriction, elongation of underlying mesoderm, etc.). Notably, genetic mutation of PCP genes in zebrafish also elicits similar spectrum of cell shape and orientation defects (*Roszko et al., 2015*). Finally, disruption of Prickle function by expression of the dominant-negative Pk2-ΔPΔL elicited the same spectrum of defects (*Figure 4C,D*, pink; *Figure 4—figure supplement 1*). Together with our data on PCP protein localization (*Figure 1*, above), these results establish the *Xenopus* neural plate as an effective platform with which to probe the relationship between epithelial cell intercalation behaviors and core PCP protein dynamics.

## Prickle2 and Vangl2 are dynamically enriched specifically at shrinking cell-cell junctions

With these imaging and analysis systems in place, we performed time-lapse imaging with an eye toward understanding the dynamic relationship between PCP protein localization and epithelial cell behaviors. This analysis revealed several novel insights. First, we noted that the accumulation of Pk2 was junction-specific, displaying overt changes in intensity precisely at tricellular junctions (*Figure 5A*). Moreover, we found that GFP-Pk2 was dynamically enriched at shrinking junctions but depleted from elongating junctions (*Figure 5A*), suggesting that dynamic enrichment of Pk2 might not simply reflect the junction's spatial orientation (e.g. to V- vs. T-junctions). This notion was support by the observation that even when adjacent junctions share a similar orientation, the GFP-Pk2 intensity at these junctions frequently differed substantially (*Videos 4* and *5*). For example, the still images in *Figure 5B* show three adjacent junctions sharing a roughly similar mediolateral alignment; none deviates by more than 20 degrees from the mediolateral at any point in the movie. Nonetheless, two of the junctions shrink during the movie (red brackets) while the third one grows (yellow brackets). Even among these similarly oriented junctions, shrink-

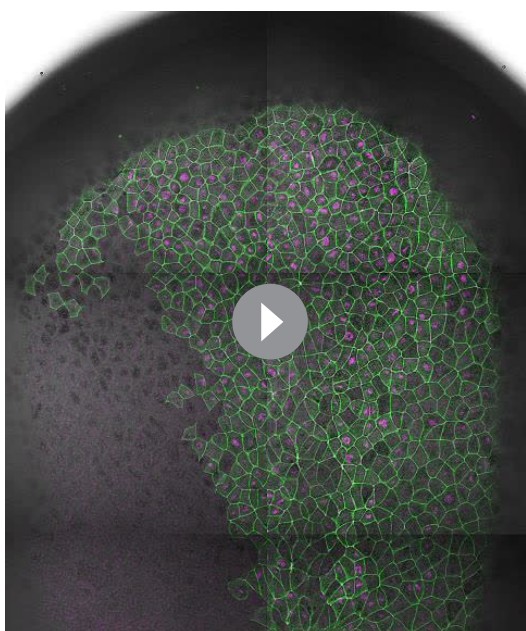

**Video 2.** Time-lapse confocal images of the dorsal side of a *Xenopus laevis* embryo from stages 12 to 16. Cells labeled with membraneGFP and nuclear H2B-RFP are merged with the DIC image of the same embryo. Still images and scale are shown in *Figure 2B*.
DOI: https://doi.org/10.7554/eLife.36456.009

ing is associated with increasing levels of Pk2-GFP intensity, and elongation with decreasing Pk2-GFP (*Figure 5C*). Similar results were obtained for Vangl2 (*Figure 5—figure supplement 1*).

To quantify these observations, we plotted the changes in GFP-Pk2 fluorescence intensity against corresponding changes in junction length, and we observed a strong and highly significant correlation (*Figure 6A*). Similar results were observed with Vangl2 (*Figure 6B*). We considered the possibility that these changes in intensity could reflect density, increasing simply if the amount of protein on a junction remains constant as that junction shrinks. To explore this idea, we examined a generic membrane marker (FP-caax) and found that while it did show a tendency to increase as junctions shrink, this increase was modest and the correlation was far weaker than that observed for PCP proteins (*Figure 6C*). Importantly, even when normalized against the membrane marker to account for changes in membrane-FP in the same junctions, the intensities of GFP-Vangl2 and GFP-Prickle2 still displayed strong and significant correlations with junction shrinkage (*Figure 6—figure supplement 1*).

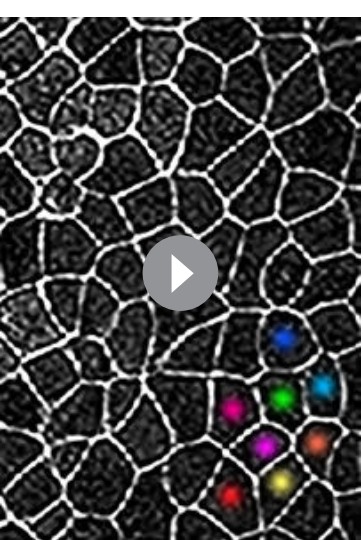

**Video 3.** Time-lapse confocal images of the dorsal side of a *Xenopus laevis* embryo from stages 12 to 16. Cells labeled with membraneGFP and pseudo-colored to help track cell rearrangements during neural convergent extension. Still images and scale are shown in *Figure 2C*.
DOI: https://doi.org/10.7554/eLife.36456.010

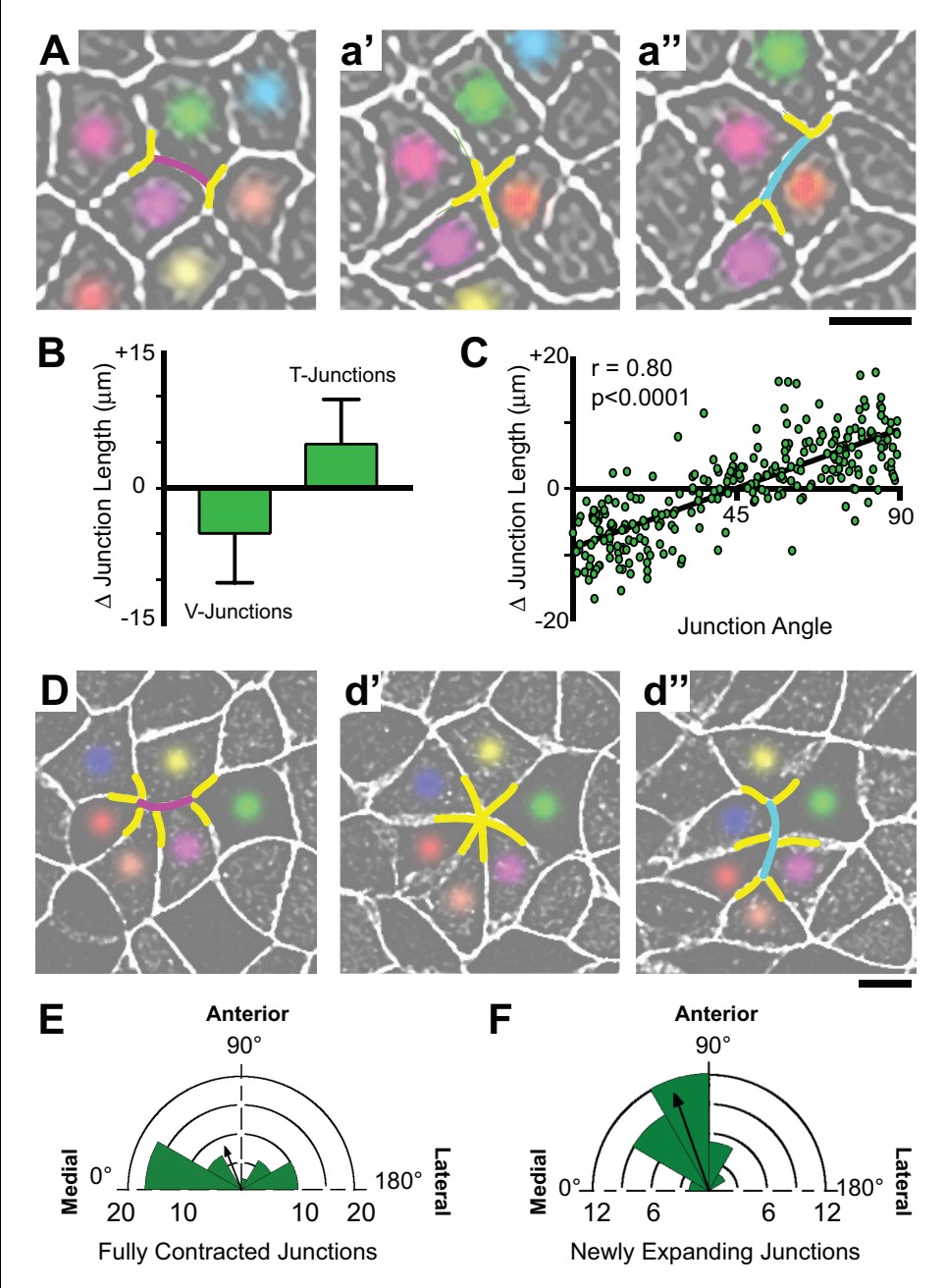

**Figure 3.** Polarized apical junction dynamics facilitate mediolateral cell intercalations. (**A**) Confocal images of junction dynamics in the *Xenopus* neural plate epithelium labeled with membraneGFP. Magenta lines mark the shrinking of a V-junction during a T transition; after complete shrinkage mediolaterally (T1-T2) (**a'**), a new junction (cyan) elongates perpendicularly along the AP axis (T2-T3) (**a''**). Scale = approx. 20 μm. (**B**) Graph showing the mean change (±s.d.) in junction length for V- and T-junctions. (**C**) Plot of the average angles of junctions over 1800 s against the change in junction length. Each dot represents on cell-cell junction. n = 267 junctions from three embryos across three different experiments. (**D**) The simultaneous mediolateral shrinking of two neighboring v-junctions (magenta) leads to formation of a multicellular rosette (**d'**), and new junctions (cyan) that emerge from the resolving rosette are oriented along the AP axis (**d''**). Scale = approx. 20 μm. (**E**) Rose diagram plotting the orientation of shrinking junctions that lead to the formation of multicellular rosettes and the mean resultant vector (arrow). n = 42 junctions from four embryos across three separate experiments. (**F**) Rose diagram plotting the orientation of new junctions emerging from resolving rosettes and mean resultant vector (arrow). n = 36 new junctions from 3 embryos across three separate experiments.

DOI: https://doi.org/10.7554/eLife.36456.011

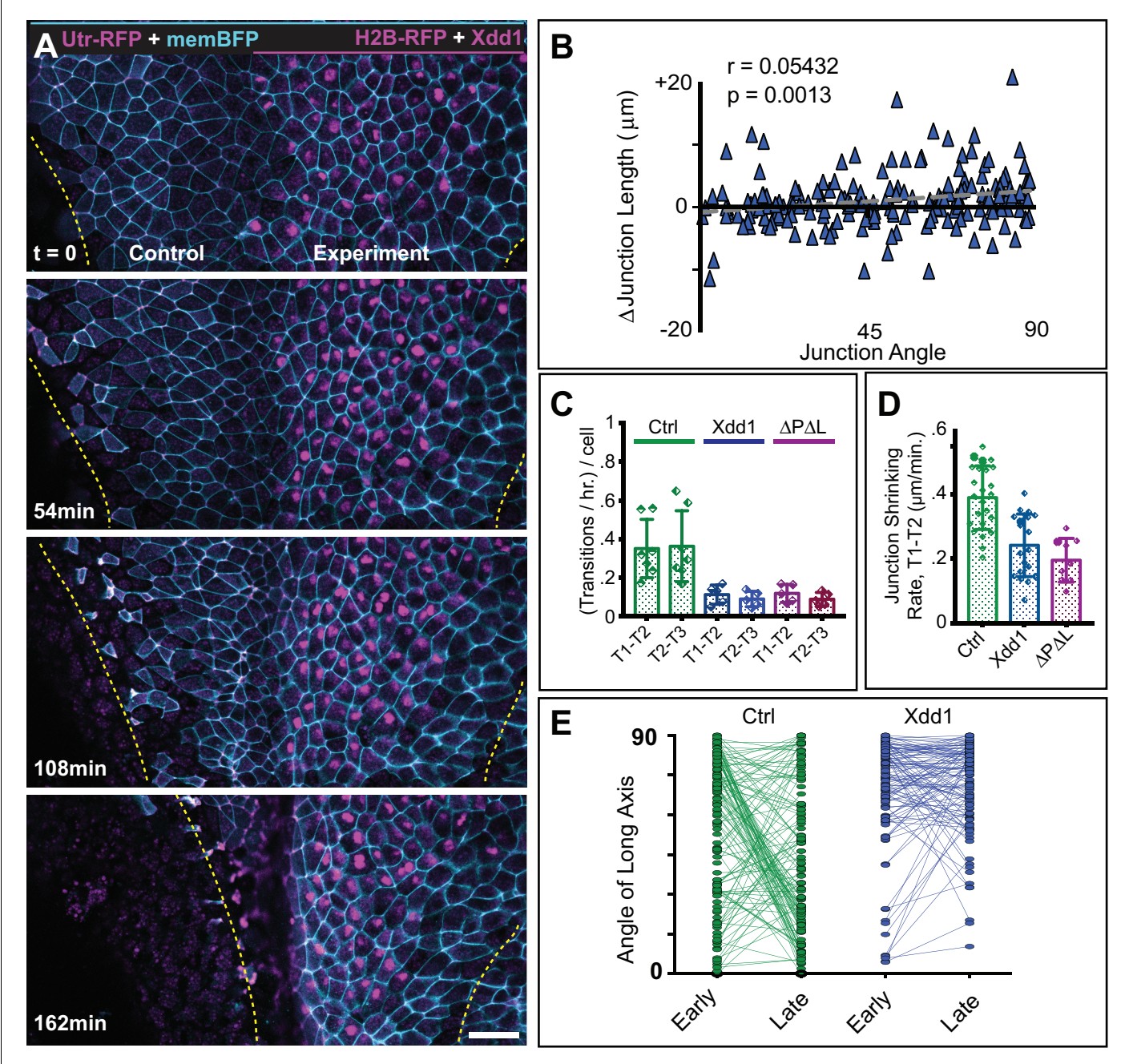

**Figure 4.** PCP function is required for polarized junction shrinking in the neural plate. (A) Confocal images from a *Xenopus* neural plate evenly labeled with membraneBFP and Utrophin-RFP (actin biosensor), but mosaically co-expressing H2B-RFP (magenta nuclei) together with Xdd1 on one side (right). The lateral boundaries of the neural plate are marked in each frame by yellow lines, demonstrating that Xdd1 expression disrupts the medial movement of the right neural fold in comparison to the control fold on the left. Scale = 50 μm (B) Graph of average angle of junctions versus junction length change for cells expressing Xdd1 to compare with control plot in *Figure 3C*. n = 187 junctions from four embryos across three experiments. (C) Graph with the total number of T transition events expressed as transitions per hour per the number of cells examined, with each point representing a single embryo. Error bars represent standard deviation. For statistical analysis, Control vs. Xdd1, and Control vs. Pk2-ΔPΔL ML, p=0.0025** for both classes of transitions (Mann-Whitney Test for significance). n = 4 experiments, seven embryos, 1167 cells (Control); three experiments, five embryos, 560 cells (Xdd1); three experiments, five embryos, 841 cells (Pk2-ΔPΔL). (D) The calculated rate of junction contraction for completed Type one to Type two transition (T1-T2) (complete contraction of a V-junction, see *Figure 3A–a'*). Error bars represent standard deviation. Ctrl vs. Xdd1, p<0.0001****, Ctrl vs. Pk2-ΔPΔL, p<0.0001****, Xdd1 vs. Pk2-ΔPΔL, p=0.2051. (Mann-Whitney statistical test). n = 24 junctions from four embryos across three experiments (Ctrl), n = 19, 2, 2 (Xdd1), and n = 9, 3, 3 (Pk2-ΔPΔL). (E) Plot of the angle of the long axis of control and Xdd1-expressing cells at Stages 12–12.5 (early) and Stages 13–14 (late), with lines connecting angles of the same cell at the two different time points. All measurements are from different regions of

*Figure 4 continued on next page*

*Figure 4 continued*

embryos that mosaically express Xdd1, similar to as shown in (A). n = 136 control cells and 114 Xdd1-expressing cells from four embryos across three experiments.

DOI: https://doi.org/10.7554/eLife.36456.012

The following figure supplement is available for figure 4:

**Figure supplement 1.** A dominant negative Pk2 disrupts polarized cell rearrangements.

DOI: https://doi.org/10.7554/eLife.36456.013

These findings suggest that the strength of asymmetric Pk2 and Vangl2 enrichment at a particular junction is at least as strongly tied to the dynamic behavior of that junction as it is to the junction's orientation. As a final test of this idea, we selected a subset of V-junctions which remained aligned within 30 degrees of the ML axis for the entire movie and then plotted the average velocity of shrinking or growth against the average intensity of GFP-Pk2 at that junction. Again, we found a highly significant correlation (*Figure 6—figure supplement 2*). Together, these data suggest that the enrichment of Pk2 and Vangl2 is governed both by the reciprocal interaction of each individual pair of neighboring cells and is intricately linked to the behavior of the shared junction between them.

## Turnover of Prickle2 and Vangl2 at cell-cell junctions is planar polarized during cell intercalation

Of the PCP protein dynamics we observed, we felt that the enrichment specifically at shrinking V-junctions was the most significant. Our data with Caax-GFP (*Figure 6C*) suggest that simply reducing the junction length is not sufficient to increase density to the full extent observed for Pk2 and Vangl2 at shrinking junctions. We reasoned, therefore, that PCP enrichment could also involve an active process, for example in which regulated turnover kinetics are more dynamic at some junctions and less so at others. FRAP studies have demonstrated that polarized PCP protein turnover is planar polarized in other cell types (*Butler and Wallingford, 2015*; *Chien et al., 2015*; *Shi et al., 2016*; *Strutt et al., 2011*), so we used this method to assess localization dynamics during cell intercalation in the closing neural tube (*Figure 7A*).

Both Pk2 and Vangl2 displayed striking differences in turnover kinetics at shrinking versus non-shrinking junctions, with significantly less recovery (i.e. higher stable fraction) for both Vangl2 and Prickle2 at shrinking junctions compared to non-shrinking junctions (*Figure 7B,C*). Moreover, when we integrated these FRAP data with time-lapse analysis of cell behaviors, we found that the stable fraction of both proteins correlated significantly with changes in junction length: more rapidly shrinking junctions displayed higher stable fractions of junctional PCP proteins (*Figure 7D,E*). Thus, the overall accumulation of Prickle2 and Vangl2 at shrinking junctions parallels the increased stability of these proteins at these sites, and together, these data suggest a key role for PCP protein trafficking in the coordination of cell-cell junction shrinkage.

## Planar polarization of actomyosin during junction shrinking in the neural epithelium is PCP-dependent

We next sought to understand the link between PCP protein localization and the actomyosin machinery known to drive cell-cell junction shrinkage. Time-lapse imaging in *Drosophila* first demonstrated that cell intercalation by junction shrinkage is accompanied by pulsed accumulations of actomyosin at V-junctions (*Bertet et al., 2004*; *Fernandez-Gonzalez et al., 2009*; *Rauzi et al., 2008*). While this process is independent of PCP signaling in *Drosophila*, similar actomyosin pulses have been observed during PCP-dependent junction shrinking in mesenchymal cells of the *Xenopus* gastrula mesoderm (*Shindo and Wallingford, 2014*). Static analyses of *Xenopus*, chicks and mice also indicate that actomyosin is enriched at mediolaterally oriented junctions (*McGreevy et al., 2015*; *Nishimura et al., 2012*; *Williams et al., 2014*). However, the spatiotemporal relationship between actomyosin dynamics and subcellular behaviors during cell intercalation in the vertebrate neural tube remains poorly defined.

Using a GFP-fusion to the myosin regulatory light chain Myl9 (*Shindo and Wallingford, 2014*), we observed a strong enrichment at V-junctions as compared to T-junctions. This enrichment was

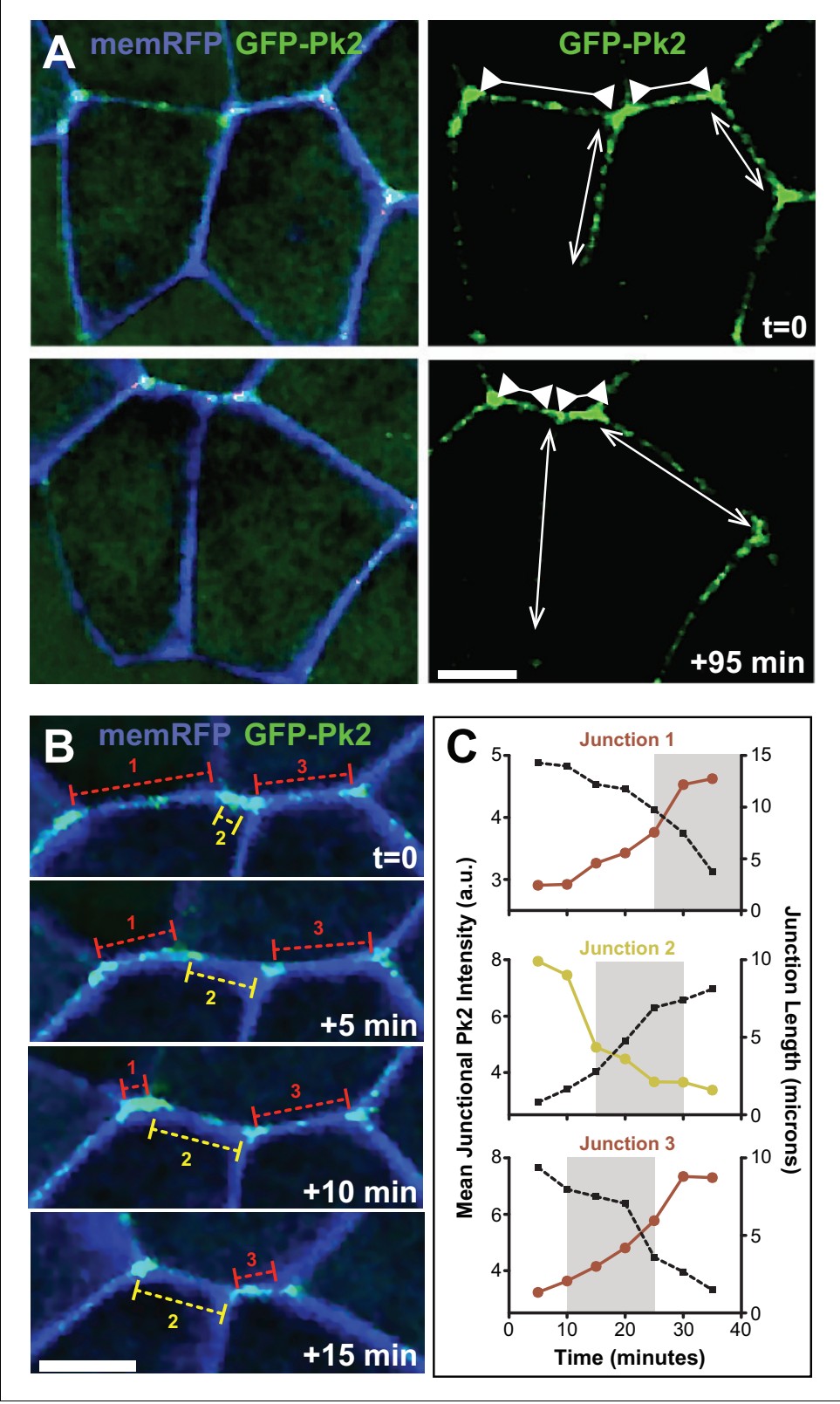

**Figure 5.** Pk2 is dynamically enriched at shrinking V-junctions. (**A**) Confocal images of neural epithelial cells labeled with membraneRFP (pseudocolored blue) and GFP-Pk2 showing the change in length of shrinking junctions (inward facing arrowheads) and growing junctions (outward facing arrows) along with the corresponding change in GFP-Pk2 intensity at two different time points. Scale = 10 µm (**B**) Higher magnification view of the

*Figure 5 continued on next page*

*Figure 5 continued*

horizontal junctions shown in A, shown at 5-min intervals. Red brackets indicated shrinking junctions; yellow brackets indicate growing junctions. Scale = 10 μm (C) Plots of intensity (red/yellow traces) and length (black dashed traces) for each of the indicated junctions in Panel B. Plots show 30 min of junction dynamics ending in either junction resolution or junction formation and expansion; gray area behind select data points indicate the interval of the four frames shown in (B), which are offset due to different junctions appearing and resolving at different times.

DOI: https://doi.org/10.7554/eLife.36456.014
The following figure supplement is available for figure 5:

**Figure supplement 1.** Distinct Vangl2 dynamics at adjacent growing and shrinking junctions.
DOI: https://doi.org/10.7554/eLife.36456.015

apparent at both the population level (*Figure 8A–C*, green) and in the significant correlation between Myl9 intensity and junction angle for individual junctions (*Figure 8D*, green, n > 450). Consistent with previous reports in other systems (*Bertet et al., 2004*; *Shindo and Wallingford, 2014*), Myl9 intensity was elevated in shrinking junctions (*Figure 8—figure supplement 1A*). These correlations are likely to be functionally relevant, because changes in myosin enrichment strongly correlated with decreases in junction length. Similar results were obtained using the actin biosensor Utrophin-RFP (*Burkel et al., 2007*), and the changes in myosin intensity were also strongly correlated to changes in actin on individual junctions (*Figure 8—figure supplement 1*).

Expression of Xdd1 or Pk2-ΔPΔL significantly disrupted the planar polarization of myosin enrichment (*Figure 8A–D*). Interestingly, these reagents appear to act via converse mechanisms: Expression of Xdd1 elicited an elevation of myosin levels at T-junctions, while expression of Pk2-ΔPΔL elicited a reduction of myosin enrichment at V-junctions (*Figure 8C*). This result was strikingly similar to the trend observed above for GFP-Pk2 intensities in these conditions (*Figure 1B*), further suggesting that PCP and Myl9 enrichments are functionally related. Thus, disruption of PCP signaling disrupts both asymmetric PCP protein localization and the planar polarization of actomyosin contraction in the closing neural tube.

## PCP proteins and actomyosin dynamics are spatiotemporally coordinated with junction shrinkage

In light of observed spatiotemporal patterns of PCP protein localization (*Figure 8*), we more closely examined Myl9 and Pk2 localization during time-lapse movies. As expected from results in other systems, Myl9-GFP displayed a pulsatile behavior at shrinking V-junctions (*Figure 9A–B*, *Video 6*). Myl9 also pulsed at T-junctions, although such pulses tended to involve larger fluctuations and failed to persistently shrink the junction (*Figure 9C*). Strikingly, Pk2 intensity also displayed pulsatile enrichment, and moreover, changes in Pk2 were strongly correlated with similar changes in Myl9 intensity at cell-cell junctions (*Figure 9A–D*). Finally, the pulses of Pk2 and Myl9 were also strongly cross-correlated in time (*Figure 9E*). Together, these data demonstrate shared dynamic patterns of core PCP protein and actomyosin localization in space and time during cell intercalation and suggest that these systems work in close concert to drive junction shrinking in the *Xenopus* neural epithelium.

## Discussion

Here, we have used image tiling and time-lapse microscopy in *Xenopus* embryos to generate high magnification movies of the closing vertebrate neural tube. These movies allowed us to quantify core PCP protein localization and

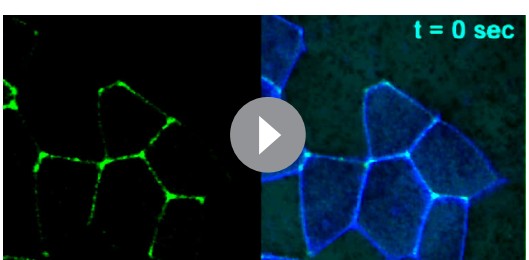

**Video 4.** Time-lapse confocal images of a *Xenopus laevis* neural plate mosaically labeled with GFP-Pk2 (left) and memRFP (right, pseudo-colored blue and merged with GFP-Pk2 and DIC channels). Still images and scale are shown in *Figure 5A and a* magnified view is shown in *Video 5* and analysis of the mediolaterally aligned junctions annotated in the beginning of the movie are provided in *Figure 5B,C*.
DOI: https://doi.org/10.7554/eLife.36456.016

Cell Biology | Developmental Biology

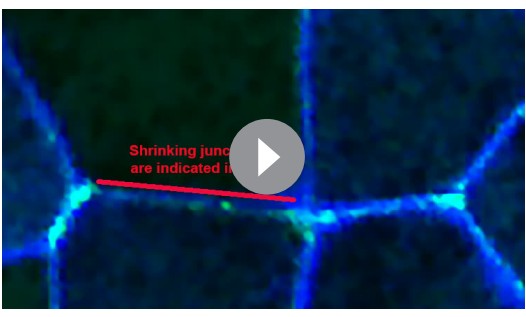

**Video 5.** Time-lapse confocal images of a *Xenopus laevis* neural plate mosaically labeled with GFP-Pk2 and memRFP (pseudo-colored blue) merged with DIC. Shrinking and growing junctions are annotated with red and yellow lines, respectively, and analysis of these junctions is provided in *Figure 5C*.
DOI: https://doi.org/10.7554/eLife.36456.017

dynamics as they relate to the cell behaviors associated with convergent extension. We focus here on junction shrinkage, which together with mediolateral protrusions is an essential sub-cellular behavior contributing to cell intercalation. We find that the Prickle2 and Vangl2 display a consistent pattern of localization and turnover in space and time during cell intercalation that is strongly linked to actomyosin assembly at cell-cell junctions. Directed, mechanistic studies will be required to fully understand the relationships reported here, but our data are nonetheless significant for providing a comprehensive and quantitative view of the spatial and temporal patterns of PCP protein localization during vertebrate collective cell movements.

## PCP protein localization in time and space during convergent extension

The spatial asymmetry of core PCP proteins is fundamental to their function. In a wide array of cell types, planar polarization is defined by Dvl and Frizzled enrichment to one region of the cell and Vangl and Prickle in a reciprocal pattern. Feedback across cell membranes is thought to reinforce initially weak asymmetry, leading to the robust asymmetry at later stages (*Butler and Wallingford, 2017*; *Strutt and Strutt, 2009*).

Interestingly, while cell intercalation was the first setting in which vertebrate PCP was defined, the first reports of asymmetric protein localization in vertebrates came instead from studies of cochlear hair cells and later from ciliated cells of the node and airway (*Rida and Chen, 2009*; *Wallingford, 2010*). Indeed, even now we know little about the spatial patterns and almost nothing about the temporal patterns of PCP protein localization in the context of convergent extension.

Currently, a consensus has emerged that PCP proteins spatially delineate an anterior-to-posterior axis in cells undergoing cell intercalation. This consensus has support from studies of various PCP proteins in diverse tissues, first in zebrafish and later in other animals (*Ciruna et al., 2006*; *McGreevy et al., 2015*; *Nishimura et al., 2012*; *Ossipova et al., 2015*; *Roszko et al., 2015*; *Yin et al., 2008*). This model is consistent with the similar pattern of anteroposterior localization for PCP proteins in the orientation of directional ciliary beating in the embryonic node and spinal cord (*Antic et al., 2010*; *Borovina et al., 2010*; *Hashimoto et al., 2010*). Moreover, this axis of polarization is consistent with embryological data suggesting that anteroposterior patterning is crucial to convergent extension in *Xenopus* (*Ninomiya et al., 2004*). However, it should be noted that several studies suggest additional regions of localization in the mediolateral ends of cells during cell intercalation (e.g. [*Jiang et al., 2005*; *Kinoshita et al., 2003*; *Panousopoulou et al., 2013*]).

Our data here are consistent with the idea that anteroposterior localization of PCP proteins is critical for cell intercalation, although they do not exclude additional roles in mediolateral protrusions, and disruption of PCP does disrupt the polarity and stability of mediolateral protrusions (*Wallingford et al., 2000*). Significantly, recent work suggests that both mediolateral protrusions and junction shrinking act together during convergent extension (*Sun et al., 2017*; *Williams et al., 2014*), so it is clear that additional studies will be required.

In our view, the more important findings here relate to the temporal aspects of PCP protein localization, as previous studies provided only static snapshots of what is a highly dynamic process. Our time-lapse studies not only reveal that Prickle2 and Vangl2 are dynamically enriched at shrinking junctions, but also suggest that the shrinking/growing status of a junction may be as important a determinant of PCP protein enrichment as is its orientation along the mediolateral/anteroposterior axes. In addition, our FRAP data reveal that turnover kinetics also differ with the dynamic behavior of the junctions, with higher stable fractions associated with shrinking junctions. Thus, our study reveals a new complexity to the pattern of PCP protein localization that likely reflects the dynamic nature of the cells involved, as they are constantly exchanging neighbors as they intercalate.

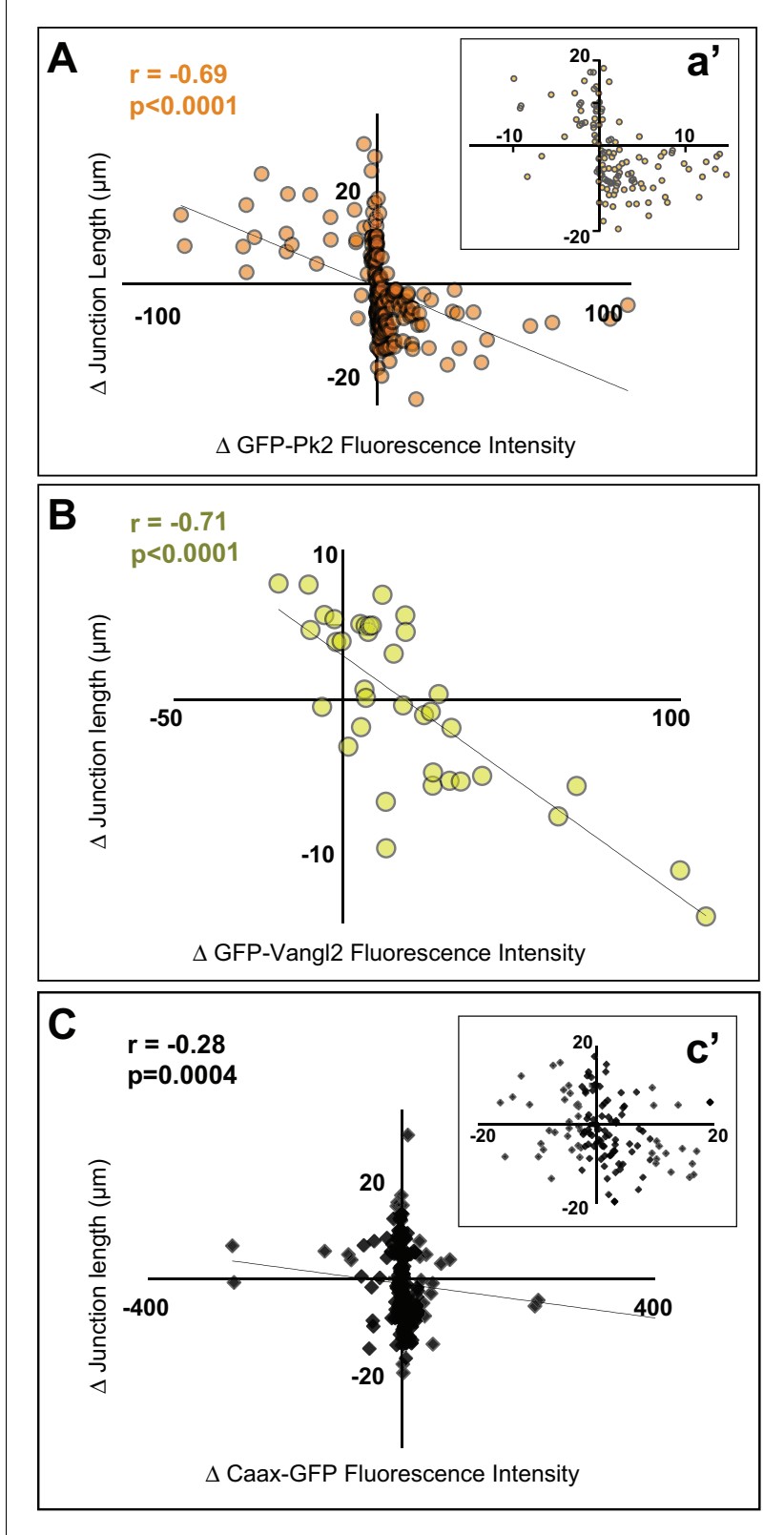

**Figure 6.** Pk2 and Vangl2 dynamics at shrinking and growing junctions. (**A**) Raw GFP-Pk2 pixel intensities strongly correlate with junction length changes. Inset shows a magnified view of the core of the plot. n = 71 junctions from five embryos across four experiments. (**B**) Raw GFP-Vangl2 pixel intensities strongly correlate with junction length changes. n = 37 junctions from 2 embryos from two different experiments. (**C**) GFP-caax displays only a weak
*Figure 6 continued on next page*

*Figure 6 continued*

correlation with junction length changes. Inset shows a magnified view of the core of the plot. n = 108 junctions from seven embryos across six experiments. Even when normalized against GFP-caax, Pk2 and Vangl2 levels show a strong correlation to junction length changes, as shown in Figure 6—figure supplement 1 *Figure 6*
DOI: https://doi.org/10.7554/eLife.36456.018

The following figure supplements are available for figure 6:

**Figure supplement 1.** PCP protein localization is correlated with junctional behavior.
DOI: https://doi.org/10.7554/eLife.36456.019

**Figure supplement 2.** Pk2 is more highly enriched at shrinking mediolateral junctions.
DOI: https://doi.org/10.7554/eLife.36456.020

Finally, our data provide an interesting complement to previous studies linking turnover at cell junctions to planar polarized PCP protein localization. In *Drosophila*, the enriched regions of PCP protein localization display higher stable fractions than do non-enriched regions, and the loss of

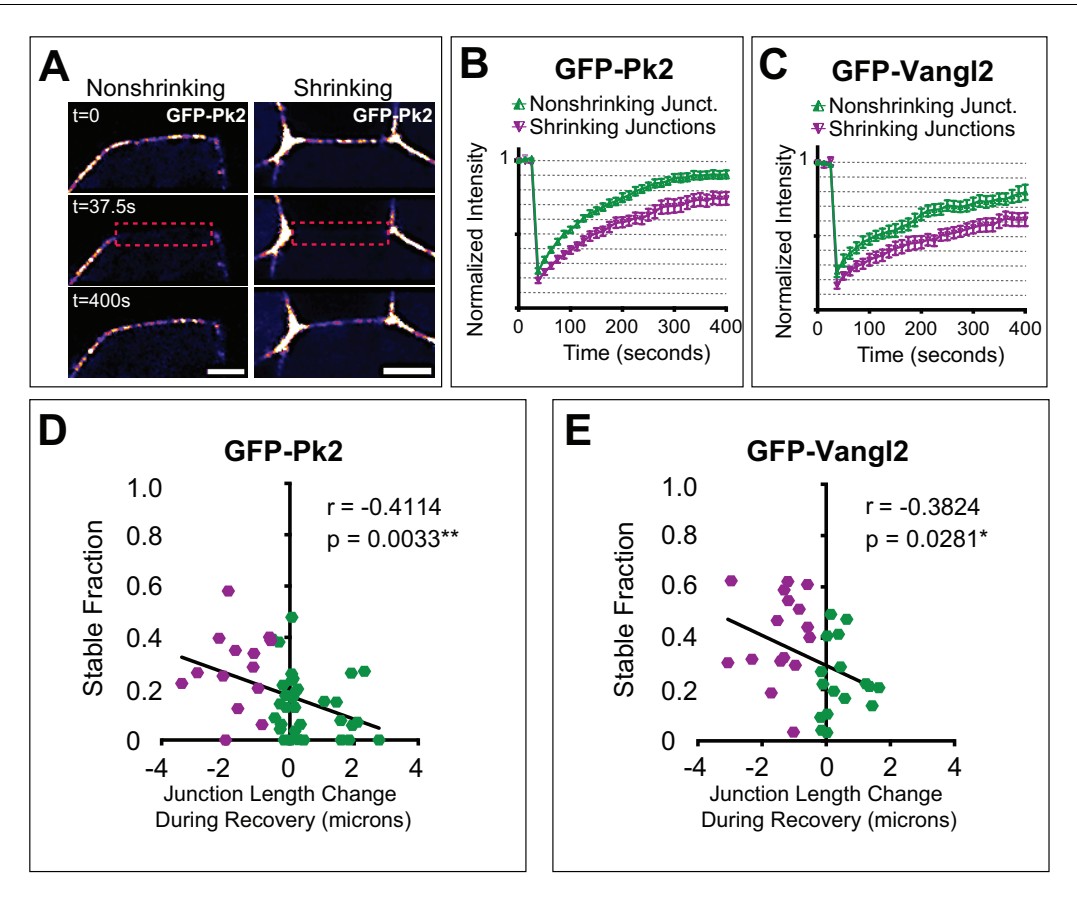

**Figure 7.** Turnover of Pk2 and Vangl2 correlates with junction behavior. (**A**) Still images from time-lapse movies captured before and after photobleaching a nonshrinking and shrinking junction of cells mosaically labeled with GFP-Pk2 in the neural plate, with a LUT applied for warmer colors representing higher fluorescence intensities. Dashed red box marks the bleached region of interest. Note that the cell on the anterior side of the junctions is unlabeled. Scale = 5 μm. (**B, C**) Graphs showing mean fluorescence recovery after photobleaching at shrinking and nonshrinking junctions for GFP-Pk2 (n = 34 nonshrinking, n = 15 shrinking) and GFP-Vangl2 (n = 17 nonshriking, n = 16 shrinking). Shrinking junctions were defined as those that were reduced by 0.5 μm or more in length over the course of bleaching and fluorescence intensity recovery. Error bars represent SEM. (**D–E**) Graphs plotting the change in junction length during photobleaching and recovery against the calculated nonmobile fraction for the individual junctions analyzed in (**B**) and (**C**) with associated linear regression model and correlation analysis statistics included. n = 49 (GFP-Prickle2); n = 33 (GFP-Vangl2).
DOI: https://doi.org/10.7554/eLife.36456.021

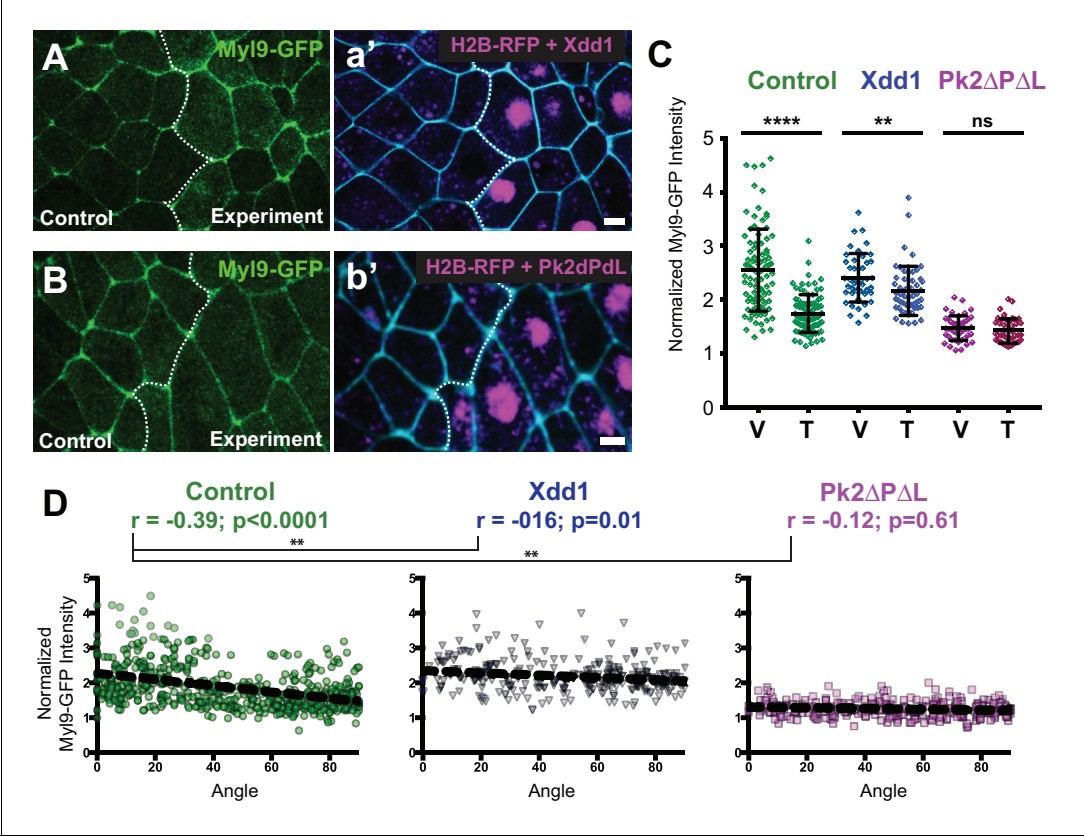

**Figure 8.** PCP function is required for polarization of actomyosin contractility during junction shrinking. (A–B). Confocal images of *Xenopus* neural epithelia labeled evenly with Myl9-GFP and membraneBFP and mosaically with H2B-RFP serving as a tracer for either Xdd1 (A) or Pk2-ΔPΔL (B) expression. Scale = 10 μm (C) Graph plotting Myl9-GFP intensity along V-junctions (0–45° relative to mediolateral axis) and T-junctions (46–90° relative to mediolateral axis) normalized as a ratio to the mean cytoplasmic intensity of the cells sharing the junction. Control cells (n = 91 V, 91T) and cells expressing Xdd1 (n = 44 V, 53 T) and Pk2-ΔPΔL (n = 45 V, 45 T). Ctrl V vs. T, p<0.0001****; Pk2-ΔPΔL V vs. T, p=0.2304; Xdd1 V vs. T, p=0.0022**; Control V vs. Xdd1 V, p=0.5826; Control T vs. Xdd1 T, p<0.0001****; Control T vs. Pk2-ΔPΔL T, p<0.0001**** (Mann-Whitney Test for significance). Error bars represent standard deviation. (D) Distributions of normalized Myl9-GFP intensity plotted against the angle of the junction at which intensity was measured in control cells and cells expressing Xdd1 or Pk2-ΔPΔL. Correlation coefficients for Xdd1 and Pk2-ΔPΔL were shown to be significantly different from controls using the Fischer R-to-Z transformation. n = 498 junctions (Control), n = 263 junctions (Xdd1) and n = 245 junctions (Pk2-ΔPΔL) from four experiments, five embryos (Xdd1); three experiments, seven embryos (Pk2-ΔPΔL).

DOI: https://doi.org/10.7554/eLife.36456.022

The following figure supplement is available for figure 8:

**Figure supplement 1.**

DOI: https://doi.org/10.7554/eLife.36456.023

asymmetric protein localization after disrupting PCP signaling is accompanied by a loss of this asymmetric turnover (*Strutt et al., 2011*). Similar results have been obtained in cells of the *Xenopus* epidermis and in the mouse oviduct (*Butler and Wallingford, 2015*; *Chien et al., 2015*; *Shi et al., 2016*). However, cell-cell neighbor exchange is minimal in those contexts, in contrast to CE, where such neighbor exchanges are constant. We therefore find it interesting that we observed a similar trend during cell intercalation, where enriched regions of PCP protein localization also display higher stable fractions of PCP protein even as these junctions shrink dramatically. Interestingly, when we calculated the mean stable fraction for PCP proteins (i.e. averaging all junctions), this value was substantially lower than what we previously observed using similar methods in *Xenopus* multiciliated cells (*Butler and Wallingford, 2015*); this difference may represent an adaptation to the dynamic nature of the junctions involved. Together with the previous studies, our data demonstrate that planar polarized junction turnover kinetics are a general feature of PCP protein localization, spanning a broad spectrum of organisms and cell types.

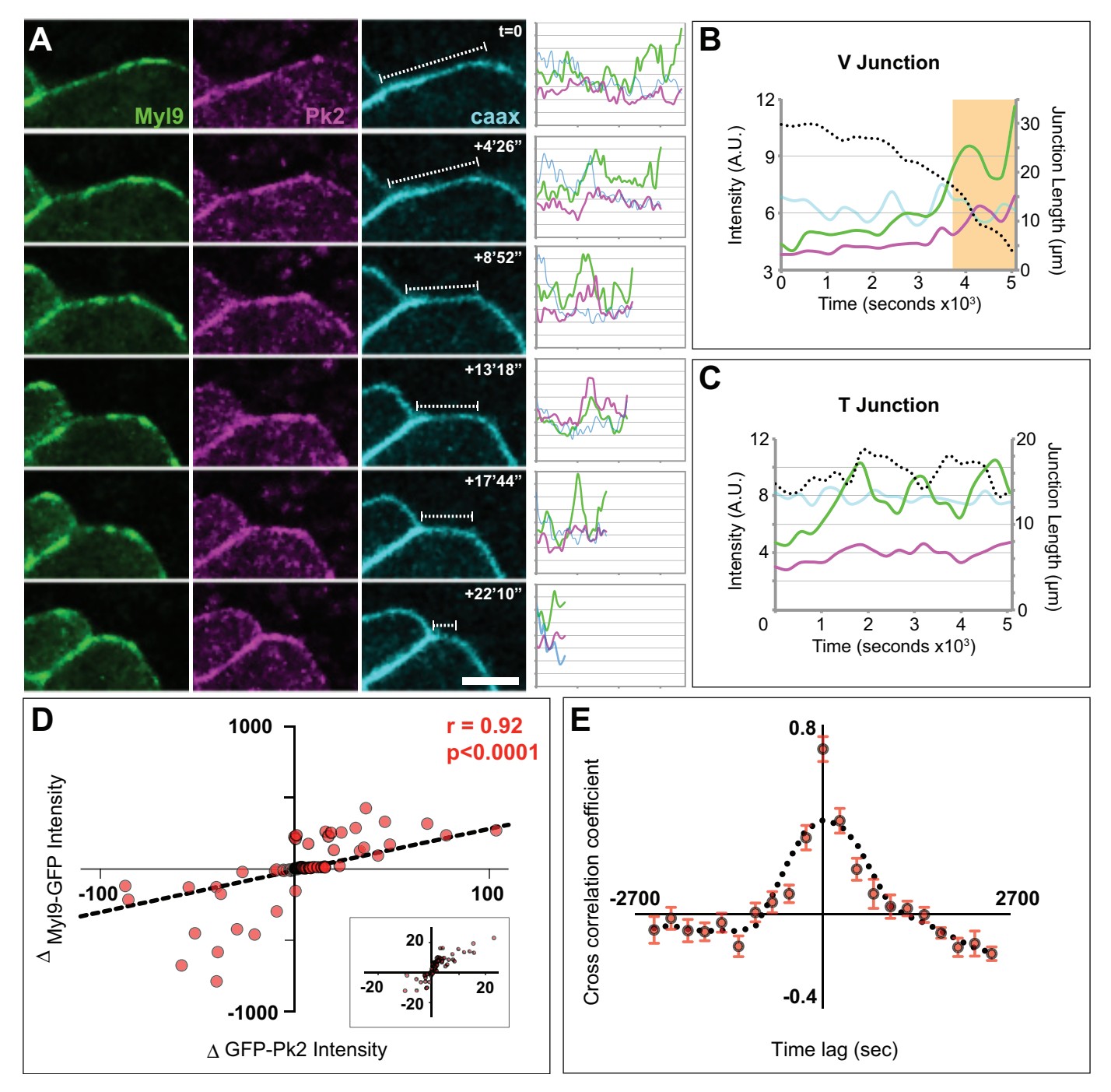

**Figure 9.** Spatiotemporal coordination of Prickle2 and actomyosin accumulation at shrinking junctions. (**A**) Confocal images of a shrinking V-junction mosaically labeled with Myl9-GFP, RFP-Pk2, and membraneBFP over the course of a 1600 s time lapse and associated intensity plot profile for each fluorophore across the length of the junction. Scale = 10 μm. (**B**) Plot of mean intensities over time for the V-junction in (**A**) which shows the pulsed co-accumulation of Pk2 (pink) and Myl9 (green) as junction length shrinks (black dashed line). The orange box marks the time points portrayed in (**A**) (**C**) Plot of mean intensities over time for a mosaically labeled T-junction showing the pulsed co-accumulation of Pk2 (pink) and Myl9 (green) as junction fluctuates between shrinking and growing (black dashed line). (**D**) Scatter plot of the change in RFP-Pk2 intensity against the change in GFP-Myl9 intensity and associated correlation; a magnified view of the core of this plot is shown in Panel c'. n = 96 junctions from three embryos across two experiments. (**E**) Cross correlation analysis of changes in RFP-Pk2 and GFP-Myl9 intensities over time, with a mean cross correlation coefficient of 0.7 at 0 s time lag demonstrating synchronous accumulation dynamics. n = 22 junctions from three embryos; error bars represent SEM.

DOI: https://doi.org/10.7554/eLife.36456.024

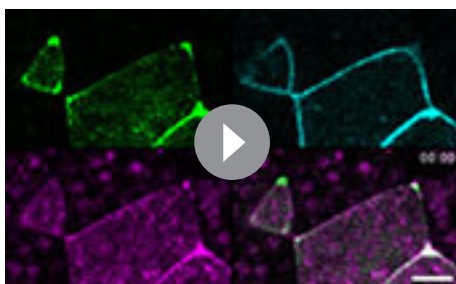

**Video 6.** Time-lapse confocal images of a *Xenopus laevis* neural plate mosaically labeled with Myl9-GFP (top left), RFP-Pk2 (bottom left), and memBFP (top right). The bottom right panel is a merge of the Myl9-GFP and RFP-Pk2 channels. Still images and analysis of these movies is provided in *Figure 9A,B*.
DOI: https://doi.org/10.7554/eLife.36456.025

## PCP protein and myosin interplay during convergent extension

Another interesting result in this study is the tight spatiotemporal relationship between PCP proteins and myosin at shrinking junctions. PCP proteins have been shown to be required for the phosphorylation of Myosin during cell intercalation in *Xenopus*, chicks, and mice (*Lienkamp et al., 2012*; *McGreevy et al., 2015*; *Nishimura et al., 2012*; *Shindo and Wallingford, 2014*; *Williams et al., 2014*), as well as in the cochlea of mice (*Lee et al., 2012*). The mechanism by which PCP proteins act on Myosin remains unclear, but of the PCP proteins, it is Dvl that is most directly implicated. Dvl acts via the formin Daam1, the PDZ-RhoGEF, and RhoA to activate Rho Kinase (*Habas et al., 2001*; *Nishimura et al., 2012*), which in turn is essential for cell intercalation (*Marlow et al., 2002*; *Ybot-Gonzalez et al., 2007*). Thus, our focus here on Prickle and Vangl (necessitated by the complexity of Dvl/Fzd function) is a limitation, because the mechanisms by which Pk2 and Vangl2 impact myosin activation are unknown.

Because mechanical feedback contributes to the oscillations of actomyosin at shrinking junctions during PCP-independent cell intercalation in *Drosophila* (e.g. [*Fernandez-Gonzalez et al., 2009*]), one attractive hypothesis involves Dvl/RhoA-mediated myosin accumulation on one cell face resulting in mechanically induced myosin accumulation on the opposing cell face. In this case, Pk2 and Vangl2 function only to ensure Dvl/Fzd localization on the other side of the junction. Alternatively, Pk2 and Vangl2 may also interact directly with actomyosin machinery by a yet to be described mechanism.

Regardless of precise mechanism, it is clear that a functional PCP system is required to drive myosin contraction, so it is interesting that two recent studies suggest that, conversely, myosin is required for normal PCP protein localization. Disruption of myosin action disrupts the polarized localization of Vangl2 in the *Xenopus* neural plate and of Pk in the ascidian notochord (*Newman-Smith et al., 2015*; *Ossipova et al., 2015*). These data may suggest a 'reciprocal' relationship between PCP proteins and myosin, an idea supported by our observations here of very tight temporal and spatial correlation between Myosin levels and Pk2 levels at cell-cell junctions. However, because there is evidence that mechanical cues can impact PCP signaling in *Drosophila*, *Xenopus*, and mice (*Aigouy et al., 2010*; *Bosveld et al., 2012*; *Chien et al., 2015*; *Luxenburg et al., 2015*), it is important to consider that both the neural plate and the notochord are engaged in large-scale collective cell movement. The broad application of myosin inhibitors would clearly disrupt global patterns of cell movement and even the orientation and magnitude of tension exerted on cell-cell junctions, so myosin inhibition might disrupt PCP protein localization secondarily. Future experiments using acute, spatially resolved myosin inhibition will be required to adequately address this issue.

## Conclusions

In sum, the data here provide a quantitative, dynamic view of PCP protein localization as it relates to a subcellular behavior that drives cell intercalation. This work provides new insights into the general problem of how developmental signaling systems such as PCP interface with fundamental cellular machines such as actomyosin. Finally, because defects in PCP signaling are strongly linked to human neural tube defects, this work also lays a foundation for understanding the growing spectrum of human disease-associated mutations in PCP genes.

## Materials and methods

### *Xenopus* manipulations

Eggs were and externally fertilized according to standard protocols (*Sive et al., 2000*). The jelly coat was removed from embryos at the two-cell stage by bathing in a solution of 2% cysteine (pH 7.9). The embryos were then washed in 1/3x Marc's Modified Ringer's (MMR) solution and microinjected in a solution of 1/3x MMR with 2% Ficoll using an Oxford manipulator. The mRNAs coding for fluorescent protein fusions were synthesized using mMessage mMachine kits (Ambion) and injected into one of eight dorsal blastomeres for even labeling of the neural plate at the following concentrations: 50 pg membraneGFP, 60 pg of membraneRFP, 80 pg of membraneBFP, 200 pg for GFP- or RFP-Pk2, 60 pg for GFP-Vangl2, 50 pg for H2B-RFP, and 30 pg for Myl9-GFP. For mosaically labeled tissues, mRNAs were injected at the 16- or 32 cell stages with approximately 70% of the totals amounts listed above. Dominant-negative Pk2 (Pk2-ΔPETΔLIM) was injected at 700–800 pg for overexpression, at the eight-cell stage, as was the dominant-negative Dvl (Xdd1), and both were similarly reduced by 70% for later stage injections. For Pk2 morpholino treatments, 20–25 ng was injected into one cell at the eight-cell stage, and 400 pg of GFP-Pk2 were used to rescue the morpholino phenotypes. Developmental stages were determined according to (*Nieuwkoop and Faber, 1994*).

### Live imaging and image quantification

Confocal imaging was done using live embryos submerged in 1/3x MMR in AttoFluor Cell Chambers (Life Technologies A7816) and between coverglasses, using silicon grease as an adhesive spacer, and carried out with a Zeiss LSM700 confocal microscope. Images were processed with the Fiji distribution of ImageJ, Imaris (Bitplane) and Photoshop (Adobe) software suites, and figures were assembled in Illustrator (Adobe). For junction length and protein enrichment measures, lines (3–6 pixels wide, depending on image scale/zoom) were drawn over cell junctional regions (excluding the tricellular junctional vertices), while cytoplasmic measures were taken using the freehand shapes tools within the apicolateral cortical regions. Mean PCP and Myl9 fluorescence intensities along a cell-cell junction were normalized to against the membrane label in analysis of changes in PCP vs. during junction length changes (Supp. *Figure 6*) and against the average of cytoplasmic intensities of the two cells sharing a junction in the analyses when cells are labeled.(*Figures 1B* and *8C*). Statistical analyses were carried out using Prism (Graphpad) software with Mann Whitney tests for significance and Spearman non-parametric correlations. Extreme outliers with fluorescent intensities more than three standard deviations away from the mean were removed from the analysis; in all datasets such outliers were very rare (<6). These outliers likely represent non-specific contraction waves sometimes observed in the *Xenopus* neural plate. For FRAP analysis, time-lapse movies were acquired after photobleaching discrete domains of core PCP GFP fusions localization. Intensity measurements were taken in Fiji, with recordings for each time point taken individually from each frame captured at bleached regions and normalized as detailed in *Goldman and Spector (2005)*. Statistical analysis was performed in Prism (Graphpad) software with exponential decay functions. Angles of junctions shrinking to form and emerging from rosettes were measured manually in Fiji, and rose diagrams were plotted with Oriana software (Kovach Computing Services). Cross-correlation analysis was performed using a free web-based statistics calculator at www.wessa.net. Stereo time-lapse imaging was performed using a Zeiss AXIO Zoom.V16 Stereomicroscope and associated Zen software. Movies were exported from Fiji and processed in Adobe Photoshop, and cell tracking was performed with the Fiji manual tracking plug-in.

## Additional information

### Funding

| Funder | Grant reference number | Author |
|---|---|---|
| National Institute of General Medical Sciences | R01GM104853 | John B Wallingford |
| Eunice Kennedy Shriver National Institute of Child Health and Human Development | R21HD084072 | John B Wallingford |

The funders had no role in study design, data collection and interpretation, or the decision to submit the work for publication.

### Author contributions

Mitchell T Butler, Conceptualization, Data curation, Formal analysis, Validation, Investigation, Visualization, Methodology, Writing—original draft, Writing—review and editing; John B Wallingford, Conceptualization, Funding acquisition, Visualization, Project administration, Writing—review and editing

### Author ORCIDs

Mitchell T Butler http://orcid.org/0000-0002-3130-1186
John B Wallingford http://orcid.org/0000-0002-6280-8625

### Ethics

Animal experimentation: This study was performed in strict accordance with the recommendations in the Guide for the Care and Use of Laboratory Animals of the National Institutes of Health. All of the animals were handled according to approved institutional animal care and use committee (IACUC) protocols (AUP-2015-00160) of the University of Texas at Austin.

### Decision letter and Author response

Decision letter https://doi.org/10.7554/eLife.36456.028
Author response https://doi.org/10.7554/eLife.36456.029

## Additional files

### Supplementary files

• Transparent reporting form
DOI: https://doi.org/10.7554/eLife.36456.026

### Data availability

Data generated or analysed during this study are included in the manuscript and supporting files

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
