## [Decision Letter]

Thank you for sending your article entitled "Spatial and temporal PCP protein dynamics coordinate cell intercalation during neural tube closure" for peer review at *eLife*. Your article is being evaluated by three peer reviewers, one of whom is a member of our Board of Reviewing Editors, and the evaluation is being overseen by Didier Stainier as the Senior Editor.

Given the list of essential revisions, including new experiments, the editors and reviewers invite you to respond within the next two weeks with an action plan and timetable for the completion of the additional work. We plan to share your responses with the reviewers and then issue a binding recommendation.

Summary:

Butler and Wallingford report their studies of the dynamic localization of Vangl2-GFP and Prickle2-GFP core components of the Planar Cell Polarity (PCP) pathway during convergent extension movements of neuroectodermal tissue in *Xenopus*. They confirm and extend earlier studies in zebrafish, frog and mouse providing snapshots of PCP protein localization that revealed enrichment of Vangl2 and Pk at the anterior edges of mediolaterally (ML) elongated cells, consistent with asymmetric localization of PCP components shown by pioneering studies in *Drosophila*. The value of this work is in careful temporal assessment of protein localization in relation to cell behavior and in particular to junction shrinking – one of the drivers of the ML intercalation process. A major conclusion of this study is that "the strength of asymmetric Pk2 and Vangl2 enrichment at a particular junction is more strongly tied to the dynamic behavior of that junction than it is to the junction's orientation or to positional information across the tissue." In addition, they show that Pk and Vangl2 are more stable at shrinking rather than growing junctions, and PCP enrichment correlates with myosin junctional localization.

This is an important study because it provides direct evidence in a live, dynamic system what the PCP field has generally assumed to be true – that PCP asymmetry accompanies myosin recruitment to promote junction shrinking during epithelial CE. The study adds further insight by showing how PCP enrichment and stability are better correlated with junction behavior than its orientation. These findings in dynamic cellular context are consistent with those of Strutt and Strutt in *Drosophila* planar polarized and more static epithelia that junctions enriched in PCP components exhibit higher stable fractions of PCP proteins. Together, these findings underscore the conservation of the molecular mechanisms of the core PCP pathway signaling between the dynamic vertebrate tissue during gastrulation/neurulation and *Drosophila* epithelia. Whereas the manuscript is largely correlative, these are very demanding experiments and the observed correlations are significant and should be of interest to the broad scientific readership given the growing list of developmental processes in which PCP pathway is being implicated.

However, some conclusions require further support and additional experimental details are needed for the reviewers to properly interpret some of the results. Moreover, it will be important to move beyond correlations.

The full reviews are also included for your reference, as they contain detailed and useful suggestions.

Essential revisions:

1) While there is good evidence that Pk and Vangl2 enrichment is better correlated with junction shrinkage than orientation, the authors do not address whether it is tied to "positional information across the tissue", i.e. whether a junction separates AP or ML neighbors. V vs T junctions - Considering the analysis is performed on a dynamic tissue where junction angles are not constant, a method that distinguishes junctions based on whether they separate AP or ML cells would be more appropriate.

2) The temporal relationship between PCP enrichment and myosin oscillations needs better documentation.

3) Please include movies for time-lapse images.

4) The data on the dynamic behavior of Vangl and Prickle fusion proteins is compelling as there are correlations with junction shrinking, actomyosin assembly, however they are just correlations. In the text the authors in general acknowledge the limitation of this analysis as for example in the summary "suggest a complex and intimate link between the dynamic localization of core PCP proteins, actomyosin assembly, and polarized junction shrinking". However, the title "Spatial and temporal PCP protein dynamics coordinate cell intercalation during neural tube closure", indicates causal relationships that are not experimentally supported and is an overstatement of the finding in this manuscript. The Title needs to be revised.

5) Moreover, it would be important to move beyond correlations. To do this, a more mechanistic understanding of relationships between PCP protein localization, myosin localization, and junction shrinking is needed. As the authors mentioned in the discussion, PCP signaling is known to regulate Myosin activity via Daam1 and Rho Kinase. One key question is whether PCP components drive myosin contractility, contractility drives PCP localization, or both. The key experiment that would take this paper beyond correlations is to perform FRAP assays under myosin inhibition. This would test whether myosin/junction shrinkage underlies the increase in Pk and Vang stability at V-junctions, or whether their stability is upstream of myosin (and perhaps due to AP position).

6) The observed enrichment of PCP components at shrinking V-junctions is a striking observation here. The authors acknowledge that "intensity could reflect increased density due to junction shrinking" as observed for generic membrane markers. But they discard this possibility because "even when normalized against such a membrane label, intensities of Vangl2 and Prickle2 still displayed a significant correlation with junction shrinkage". So, they put forward "the alternative hypothesis" whereby the observed enrichment could be an active process, the hypothesis they pursue and find experimental support for. However, these hypotheses are not mutually exclusive. It would be important to present data for the changes in the density of the PCP and general membrane markers, to more fully understand the relative contributions of these two mechanisms. For example, it would be valuable to see a line for the intensity of the generic membrane marker on the graph in Figure 8B. And although PCP fusion protein localization is normalized to a generic membrane marker in Figure 6—figure supplement 1, the legend of Figure 1 states that fluorescence intensity was normalized to the cytoplasm. Additional figure legends should specify to what fluorescence intensity is normalized.

7) Figure 4 presents defective CE in neuroectoderm expressing Xdd1. Junction shrinking is analyzed. However, visually abnormal cell alignment of cells is evident and should be quantified. It seems that instead of ML alignment, cell bodies are aligned with the AP embryonic axis, as has been seen in some zebrafish PCP mutants (e.g. Roszko et al., 2015). If this is the case, how V junction should be defined? Simply as ML aligned or should be longer junction (as V junctions in ML-elongated cells are). This should be analyzed and considered.

8) Reduced number of productive T1 transitions is perceived as the main phenotype. However, it would be important to analyze the total frequency of any transitions and intercalations. Does Xdd1 overexpression lead to loss of polarized transitions and intercalations or does it generally impair junction shrinking and intercalations. This is very important to assess whether Xdd1 overexpression impairs CE in the same way as loss of individual PCP components in mouse or zebrafish mutants.

9) The paper could be improved by a restructuring of the discussion to present a working model of PCP and myosin regulation during cell intercalation in CE, including how the present study supports or challenges prevailing models.

Reviewer #1:

Butler and Wallingford report their studies of the dynamic localization of Vangl2-GFP and Prickle2-GFP core components of the Planar Cell Polarity (PCP) pathway during convergent extension movements of neuroectodermal tissue in *Xenopus*. They confirm and extend earlier studies in zebrafish, frog and mouse providing snapshots of PCP protein localization that revealed enrichment of Vangl2 and Pk at the anterior edges of mediolaterally (ML) elongated cells, consistent with asymmetric localization of PCP components shown by pioneering studies in *Drosophila*. The value of this work is in careful temporal assessment of protein localization in relation to cell behavior and in particular to junction shrinking – one of the drivers of the ML intercalation process. The authors propose that Vangl and Prickle are dynamically enriched at shrinking junctions and suggest "the shrinking/growing status of a junction is a better indicator of PCP protein enrichment than its orientation in the ML/AP axis. Another important insight from FRAP experiments reported here is that turnover kinetics also vary with the dynamic junction behavior such that more stable fractions are associated with shrinking junctions and this is correlated with behavior of Myosin at junctions. These findings in dynamic cellular context are consistent with those of Strutt and Strutt in *Drosophila* planar polarized and more static epithelia that junctions enriched in PCP components exhibit higher stable fractions of PCP proteins. Together, these findings underscore the conservation of the molecular mechanisms of the core PCP pathway signaling between the dynamic vertebrate tissue during gastrulation/neurulation and *Drosophila* epithelia. Whereas the manuscript is largely correlative, these are very demanding experiments and the observed correlations are significant and should be of interest to the broad scientific readership given the growing list of developmental processes in which PCP pathway is being implicated.

However, there are several conclusions that require additional experimental and several conclusions need further discussion. Considering the following points will significantly improve the manuscript.

- The data on the dynamic behavior of Vangl and Prickle fusion proteins is compelling as there are correlations with junction shrinking, actomyosin assembly, however they are just correlations. In the text the authors in general acknowledge the limitation of this analysis as for example in the summary "suggest a complex and intimate link between the dynamic localization of core PCP proteins, actomyosin assembly, and polarized junction shrinking". However, the title "Spatial and temporal PCP protein dynamics coordinate cell intercalation during neural tube closure", indicates causal relationships that are not experimentally supported and is an overstatement of the finding in this manuscript. The title needs to be revised.

- Moreover, it would be important to move beyond correlations. To do this, a more mechanistic understanding of relationships between PCP protein localization, myosin localization, and junction shrinking is needed. As the authors mentioned in the discussion, PCP signaling is known to regulate Myosin activity via Daam1 and Rho Kinase. It would be important to know whether loss of function of either Rho or Daam1 leads to an uncoupling of Myl9 and Pk2 at V junctions. Another important question is whether PCP components drive myosin contractility, contractility drives PCP localization, or both. They show that function of Pk and Dvl is necessary for polarized Myosin localization, but is it sufficient? I.e. Does ectopic expression of PCP components drive asymmetric Myosin accumulation? And would pharmacologically increasing or decreasing myosin contractility lead to increased or decreased Pk2 localization?

- The observed enrichment of PCP components at shrinking V-junctions is a striking observation here. The authors acknowledge that "intensity could reflect increased density due to junction shrinking" as observed for generic membrane markers. But they discard this possibility because "even when normalized against such a membrane label, intensities of Vangl2 and Prickle2 still displayed a significant correlation with junction shrinkage". So, they put forward "the alternative hypothesis" whereby the observed enrichment could be an active process, the hypothesis they pursue and find experimental support for. However, these hypotheses are not mutually exclusive. It would be important to present data for the changes in the density of the PCP and general membrane markers, to more fully understand the relative contributions of these two mechanisms. For example, it would be valuable to see a line for the intensity of the generic membrane marker on the graph in Figure 8B. And although PCP fusion protein localization is normalized to a generic membrane marker in Figure 6—figure supplement 1, the legend of Figure 1 states that fluorescence intensity was normalized to the cytoplasm. Additional figure legends should specify to what fluorescence intensity is normalized.

- Expression of Xdd1 and Pk2-∆P∆L is used to disrupt PCP signaling, and Xdd1 has been used in several previous studies as a model for loss of PCP signaling. However, dominant negative approaches can have additional effects. Indeed, as the authors acknowledge these two reagents appear to act via converse mechanisms on the planar polarization of myosin enrichment. This hints that it is not simply a loss of PCP signaling that disrupts junction polarity and contractility, but rather that each of these constructs influences cell polarity in distinct ways. A closer examination of the interplay between Pk2/Vangl2, Dvl, and Myosin would distinguish between unintended effects on dominant-negative constructs and biologically relevant consequences of PCP signaling.

- It is important to note possible caveats associated with studying the behavior of misexpressed fusion proteins. This has been a standard in the field, but it is puzzling that in this work Vangl-GFP appears to be absent on posterior membranes, whereas Roszko et al., 2015) using antibodies detected endogenous Vangl2 in zebrafish mesodermal and neuroectodermal cells along the entire cell membrane and using Vangl2-GFP fusion protein along the entire cell membrane but anteriorly enriched. Do longer exposures show lower level of Vangl-GFP at other membranes as well?

- Similarly, arguable is the conclusion in the Discussion section "More important than the spatial localization of PCP proteins during cell intercalation is the temporal aspect reported here." First, as the authors note the asymmetric distribution of core PCP components and that junctions enriched in PCP components exhibit higher stable fractions of PCP proteins has been observed in more static epithelia in *Drosophila*. The more dynamic aspects reported here for shrinking junctions may represent a phase in the course of PCP signaling. Indeed, this phase likely depends on the earlier phases of PCP signaling where both in *Drosophila* and zebrafish endogenous Vangl accumulates uniformly at cell membrane and then its distribution becomes asymmetric. As disruption of PCP signaling disrupts all asymmetries including the dynamic ones reported here, the "more important" statement is not justified.

- Figure 4 presents defective CE in neuroectoderm expressing Xdd1. Junction shrinking is analyzed. However, visually abnormal cell alignment of cells is evident and should be quantified. It seems that instead of ML alignment, cell bodies are aligned with the AP embryonic axis, as has been seen in some zebrafish PCP mutants (e.g. Roszko et al., 2015). If this is the case, how V junction should be defined? Simply as ML aligned or should be longer junction (as V junctions in ML-elongated cells are). This should be analyzed and considered.

- Reduced number of productive T1 transitions is perceived as the main phenotype. However, it would be important to analyze the total frequency of any transitions and intercalations. Does Xdd1 overexpression lead to loss of polarized transitions and intercalations or does it generally impair junction shrinking and intercalations. This is very important to assess whether Xdd1 overexpression impairs CE in the same way as loss of individual PCP components in mouse or zebrafish mutants.

Reviewer #2:

Butler and Wallingford investigate PCP protein localization during convergent extension (CE) movements that accompany neural tube closure in *Xenopus* embryos. They find that Pk-GFP and Vangl2-GFP are asymmetrically distributed at intercellular junctions and that the proteins are enriched most strongly at actively shrinking junctions. A major conclusion of this study is that "the strength of asymmetric Pk2 and Vangl2 enrichment at a particular junction is more strongly tied to the dynamic behavior of that junction than it is to the junction's orientation or to positional information across the tissue." In addition, they show that Pk and Vangl2 are more stable at shrinking rather than growing junctions, and PCP enrichment correlates with myosin junctional localization.

This is an important study because it provides direct evidence in a live, dynamic system what the PCP field has generally assumed to be true – that PCP asymmetry accompanies myosin recruitment to promote junction shrinking during epithelial CE. The study adds further insight by showing how PCP enrichment and stability are better correlated with junction behavior than its orientation.

However, some conclusions require further support and additional experimental details are needed for the reviewer to properly interpret some of the results. Specifically, while the there is good evidence that Pk and Vangl2 enrichment is better correlated with junction shrinkage than orientation, they do not address whether it is tied to "positional information across the tissue", i.e. whether a junction separates AP or ML neighbors. Second, the temporal relationship between PCP enrichment and myosin oscillations needs better documentation. Third, the paper does not address whether myosin activity or AP position accounts for the observed increased Pk and Vangl2 stability. Most of these points could be addressed with existing data sets. Finally, I feel that the paper could be improved by a restructuring of the discussion to present a working model of PCP and myosin regulation during cell intercalation in CE, including how the present study supports or challenges prevailing models.

Revisions needed to strengthen the major conclusions:a) Please include movies for time-lapse images.

b) All experiments use exogenously expressed GFP constructs. Do Vangl2, Pk2 and MyI9 endogenous proteins show the same degree of polarization?

c) V vs T junctions- Considering the analysis is performed on a dynamic tissue where junction angles are not constant, a method that distinguishes junctions based on whether they separate AP or ML cells would be more appropriate. For example, in Figure 5A, an expanding junction is shown oriented roughly perpendicular to a neighboring shrinking junction. The junction changes its orientation ~30 degrees as it lengthens. When during this process would the angle of the junction be measured? Since junction angles fluctuate more rapidly than cells exchange neighbors, a more meaningful categorization of V vs T could be whether the junction separates AP or ML neighbors. Since the AP relationship between cells is thought to be the primary determinant of PCP asymmetry in other systems, I feel this is important to consider.

d) Pk and Vangl stability – The data shown that Pk and Vangl2 stability correlates with shrinking junctions, but does it also correlate with junction angle or AP neighbor relationships? Perhaps stability is dependent on whether Pk and Vangl are incorporated in an intercellular PCP complex, which would be expected to occur between AP neighbors, but less so at ML neighbors (Strutt et al., 2011).

e) Temporal correlation between Pk and myosin – Figure 8A,B. A major conclusion of the paper is that "recruitment of Prickle2 and Vangl2 to cell-cell junctions was temporally and spatially coordinated with planar polarized oscillations of actomyosin enrichment", but the temporal correlations and oscillatory behaviors are not clear from the image shown in Figure 8B or intensity plot over the corresponding junction in Figure 8B. Only one example is given, and the behavior is not obviously oscillatory. Accompanying movies and analysis of additional shrinkage events would strengthen the data and potentially provide insights into the order of events, i.e. whether Pk enrichment precedes myosin accumulation.

Clarifications to the experimental procedures and data analysis:a) Figure 1B. – A more complete description of how GFP intensities were measured is needed. Is the average GFP intensity across the junction plotted? Are junctions between two cells that both express Pk-GFP or Vangl2-GFP included in the analysis? Or are only individual GFP+ cells that are surrounded by GFP- neighbors included?

b) Figure 5A-B. Pk-GFP intensity is particularly strong at vertices, which raises the question of how the boundaries of a junction are defined. Are the intensities at vertices included or excluded as part of the junction? In both growing and shrinking junctions? These bright points could really skew the data.

c) Figure 5B. Could the authors provide more information about how the changes in GFP intensity are determined? Are they choosing the same time points used to calculate the change in junction length (Intensity at t=end minus intensity at t=0)? Or is this the net change (max minus min) over the course of junction length changes? How are the max and min junction lengths chosen?

d) Figure 5B. There are many junctions that display a substantial change in length but no change in Pk intensity. I'm not sure how to think about these data points. Can the authors comment? If Pk2/PCP localization instructs junction shrinkage, then these results are unexpected. Perhaps these data points represent unproductive shrinkage events that never result in complete junction loss and neighbor exchange? If the data were separated into productive shrinkage events that lead to junction loss vs fluctuating junctions would the correlation between Pk intensity and length changes be even stronger?

e) Figure 6A-E. In the FRAP analysis, were both V and T-junctions analyzed (oriented along the full range of 0-90 degrees)? Does Pk-GFP stability correlate with the junction angle (see point 1d above)?

Reviewer #3:

I enjoyed reading this manuscript and believe it will be a valuable addition to the literature. It is essentially 'descriptive', in the positive sense of providing a careful quantitative description of events that will be of great value in underpinning further work in the field. The findings are correlative rather than mechanistic. Looking over my brief notes I made as I read through, I also find they are largely confirmatory:

- establish/confirm Pk2 and Vangl2 colocalize preferentially on A/P cell boundaries (but asymmetry weak).

- confirm that convergent extension occurs in epithelium of neural plate via T1 transitions and rosette formation and is dependent on PCP pathway function.

- GFP-Pk2/GFP-Vangl2 are enriched on shrinking junctions (this of course follows from the known localizations on A/P cell boundaries and that the tissue undergoes convergent extension on the AP axis). Interestingly, Pk2/Vangl2 are preferentially enriched on shrinking junctions: possibly they are actively recruited to shrinking junctions, or actively driving shrinkage? Or maybe turnover is slow, so get concentrated on shrinking junctions by virtue of the shrinkage?

- find by FRAP that Pk2/Vangl2 show higher stable fraction on shrinking junctions, where their concentrations are higher. This is consistent with *Drosophila* data (e.g. Strutt et al., 2011?) suggesting at higher junctional concentrations, PCP proteins show higher stable fractions?

- see expected pulsatile actomyosin behavior on shrinking junctions which is PCP pathway dependent. Pk2 seems to pulse with myosin, due to shared membrane enrichment?

There is an interesting hint that there may be an active relationship between PCP protein behavior and actomyosin dynamics but cause and effect is not investigated.

This would sit best in a good developmental biology journal?

---

## [Author Response]

[Editors’ note: formal revisions were requested, following approval of the authors’ plan of action.]

Essential revisions:1) While there is good evidence that Pk and Vangl2 enrichment is better correlated with junction shrinkage than orientation, the authors do not address whether it is tied to "positional information across the tissue", i.e. whether a junction separates AP or ML neighbors. V vs T junctions- Considering the analysis is performed on a dynamic tissue where junction angles are not constant, a method that distinguishes junctions based on whether they separate AP or ML cells would be more appropriate.

We agree that this is an important point and we addressed it in three ways.

First, because "junction angles are not constant," we re-examined our data from Figure 3B and C. We found that the average change in junction angle during the period of analysis for this dataset was only 5 degrees (+/- 4 degrees), with no changes greater than 20 degrees observed. This result is now reported (subsection “Epithelial convergent extension in the closing *Xenopus* neural tube involves PCP-194 dependent polarized junction shrinking”), as it will help the reader to evaluate the dataset. This result argues that our use of junction orientation is an effective proxy for junctions that separate A/P neighbors.

Second, even with these data, we feel the reviewer's point is valid, so we added a new analysis using more stringent criteria. Because we feel it is essential to continue identifying junctions by their orientation (this is an unambiguous metric and is the standard in the field for many labs), we selected junctions for this analysis that remain within 30 degrees of mediolateral for the entire duration of analysis, and for these junctions, we plotted the average velocity of shrinking or growth against the average Pk2 intensity for every time point in the movie (subsection “Prickle2 and Vangl2 are dynamically enriched specifically at shrinking cell-cell 266 junctions”, new Figure 6—figure supplement 2). Again, we observed a very strong correlation.

Finally, we softened our language. We no longer claim that behavior is "better" correlated than orientation, but instead suggest that it is "at least as" correlated.

We hope these changes address the reviewers’ intended question; if not, we apologize and ask for clarification.

2) The temporal relationship between PCP enrichment and myosin oscillations needs better documentation.

We agree, and we have made three changes:

1) We have added better, higher magnification images supporting this claim, as well as intensity plots for the frames shown. These are provided in Figure 9A of the revision.

2) We provide a representative time-lapse movie.

3) We provide data showing a strong cross-correlation between Pk2 and Myl9 levels (new Figure 9E), as detailed below in our response to comment 5.

3) Please include movies for time-lapse images.

We regret this oversight. We have supplied several representative time-lapse movies to accompany the figures.

4) The data on the dynamic behavior of Vangl and Prickle fusion proteins is compelling as there are correlations with junction shrinking, actomyosin assembly, however they are just correlations. In the text the authors in general acknowledge the limitation of this analysis as for example in the summary "suggest a complex and intimate link between the dynamic localization of core PCP proteins, actomyosin assembly, and polarized junction shrinking". However, the title "Spatial and temporal PCP protein dynamics coordinate cell intercalation during neural tube closure", indicates causal relationships that are not experimentally supported and is an overstatement of the finding in this manuscript. The Title needs to be revised.

We have re-titled the manuscript: "Spatial and temporal analysis of PCP protein dynamics during neural tube closure".

5) Moreover, it would be important to move beyond correlations. To do this, a more mechanistic understanding of relationships between PCP protein localization, myosin localization, and junction shrinking is needed. As the authors mentioned in the discussion, PCP signaling is known to regulate Myosin activity via Daam1 and Rho Kinase. One key question is whether PCP components drive myosin contractility, contractility drives PCP localization, or both. The key experiment that would take this paper beyond correlations is to perform FRAP assays under myosin inhibition. This would test whether myosin/junction shrinkage underlies the increase in Pk and Vang stability at V-junctions, or whether their stability is upstream of myosin (and perhaps due to AP position).

In the requested "action plan," we proposed addressing this concern with a cross-correlation analysis of PCP protein and Myosin dynamics to ask if these pulses are truly coincident or if in fact one pulse precedes the other. We now report a strong cross-correlation between Myl9 and Pk2 intensities - which we feel adds value to the paper (New Figure 9E). However, we find that neither one clearly preceded the other in this analysis. This may reflect true simultaneous enrichment, or it may indicate that the time-resolution of our movies is insufficient to detect a difference.

We regret, then, that without substantial additional experimentation, we are unable to make more mechanistic conclusions. That being said, we respectfully point out that the data here nonetheless represent both a quantum leap forward in analysis of PCP protein localization during CE and a very thorough body of work.

We will revise the manuscript to (a) explicitly acknowledge the limitations of our descriptive approach (Discussion section) and (b) provide a careful synthesis of our current understanding of the relationships between PCP proteins and myosin in the Discussion section.

We very much hope the reviewers find this proposal acceptable.

6) The observed enrichment of PCP components at shrinking V-junctions is a striking observation here. The authors acknowledge that "intensity could reflect increased density due to junction shrinking" as observed for generic membrane markers. But they discard this possibility because "even when normalized against such a membrane label, intensities of Vangl2 and Prickle2 still displayed a significant correlation with junction shrinkage". So, they put forward "the alternative hypothesis" whereby the observed enrichment could be an active process, the hypothesis they pursue and find experimental support for. However, these hypotheses are not mutually exclusive. It would be important to present data for the changes in the density of the PCP and general membrane markers, to more fully understand the relative contributions of these two mechanisms. For example, it would be valuable to see a line for the intensity of the generic membrane marker on the graph in Figure 8B.

This is another important point, so we have moved our raw correlation data for Pk2 and Vangl2 to a new Figure 6 that also includes the correlation data for the general membrane marker, Caax-GFP. We present these raw data first and then discuss the Pk2/Vangl2 data normalized against Caax-GFP, which is now shown in Figure 6—figure supplement 1 of the revision. We also reworded how we interpret these data to reflect that reduced turnover could in fact help produce the increased enrichment we see for the PCP proteins over the membrane label rather than use language that suggests they are mutually exclusive

And although PCP fusion protein localization is normalized to a generic membrane marker in Figure 6—figure supplement 1, the legend of Figure 1 states that fluorescence intensity was normalized to the cytoplasm. Additional figure legends should specify to what fluorescence intensity is normalized.

All legends now accurately reflect if and how normalization was used for each particular dataset. We also included additional language in the methods to point out the circumstances and details for the two alternative normalization methods.

7) Figure 4 presents defective CE in neuroectoderm expressing Xdd1. Junction shrinking is analyzed. However, visually abnormal cell alignment of cells is evident and should be quantified. It seems that instead of ML alignment, cell bodies are aligned with the AP embryonic axis, as has been seen in some zebrafish PCP mutants (e.g. Roszko et al., 2015). If this is the case, how V junction should be defined? Simply as ML aligned or should be longer junction (as V junctions in ML-elongated cells are). This should be analyzed and considered.

We agree that these are important points. We have extracted such data from our movies and now provide them in the new Figure 4 and Figure 4—figure supplement 1 of the revision. This analysis revealed that, as the reviewer intuited, there is a defect in the orientation of the long axes of cells with Xdd1 or dom-neg Pk2 expression. Moreover, we find that the planar polarization of junction shrinkage is entirely randomized by Xdd1 expression, and consistent with the reduction of T1 transitions, junction shrinkage is reduced overall.

Regarding the question of how V junctions should be defined if cell elongation is changed is an interesting one. We strongly feel that it remains critical for junctions to have unambiguous definitions, so we prefer to continue using orientation to define junctions, as outlined in comment 1, above.

Importantly, our analysis suggests that junction-shrinking events are not re-oriented but rather are abrogated and randomized.

8) Reduced number of productive T1 transitions is perceived as the main phenotype. However, it would be important to analyze the total frequency of any transitions and intercalations.

We regret being unclear: The analysis shown in what is now Figure 4C and D is for all junction shrinkage events, regardless of orientation. Those data, combined with the analysis provided in comment 7, should address this concern.

Does Xdd1 overexpression lead to loss of polarized transitions and intercalations or does it generally impair junction shrinking and intercalations. This is very important to assess whether Xdd1 overexpression impairs CE in the same way as loss of individual PCP components in mouse or zebrafish mutants.

If we understand this correctly, the analysis we provide in comments 1 and 7 should address this concern.

9) The paper could be improved by a restructuring of the discussion to present a working model of PCP and myosin regulation during cell intercalation in CE, including how the present study supports or challenges prevailing models.

We have added extensively to the Discussion section, but we preferred stop short of presenting a "working model," which we feel would be premature. As mentioned above, our data do not provide mechanistic insights into this question, though we feel they do add important dynamic information about the relationship between PCP and myosin. Moreover, as one of the reviewers points out in the last minor comment below, this relationship must be handled with great care. We hope the reviewers feel our additional verbiage is sufficient to render the discussion useful.